# DENND6A links Arl8b to a Rab34/RILP/dynein complex, regulating lysosomal positioning and autophagy

Rahul Kumar [1] ✉, Maleeha Khan [1,2], Vincent Francis[1,2], Adriana Aguila [1], Gopinath Kulasekaran[1], Emily Banks [1] & Peter S. McPherson [1] ✉

Lysosomes help maintain cellular proteostasis, and defects in lysosomal positioning and function can cause disease, including neurodegenerative disorders. The spatiotemporal distribution of lysosomes is regulated by small GTPases including Rabs, which are activated by guanine nucleotide exchange factors (GEFs). DENN domain proteins are the largest family of Rab GEFs. Using a cell-based assay, we screened DENND6A, a member of the DENN domain protein family against all known Rabs and identified it as a potential GEF for 20 Rabs, including Rab34. Here, we demonstrate that DENND6A activates Rab34, which recruits a RILP/dynein complex to lysosomes, promoting lysosome retrograde transport. Further, we identify DENND6A as an effector of Arl8b, a major regulatory GTPase on lysosomes. We demonstrate that Arl8b recruits DENND6A to peripheral lysosomes to activate Rab34 and initiate retrograde transport, regulating nutrient-dependent lysosomal juxtanuclear repositioning. Loss of DENND6A impairs autophagic flux. Our findings support a model whereby Arl8b/DENND6A/Rab34-dependent lysosomal retrograde trafficking controls autophagy.

Lysosomes are heterogeneous and highly dynamic organelles[1]. There has been a paradigm shift in the perception of lysosomes from solely degradative organelles to a multifunctional cellular compartment with roles in immune defense, repair of damaged plasma membranes, release of exosomes, regulation of cell adhesion and invasion, programmed cell death, metabolic signaling, and control of gene expression[2]. Since lysosomes, endolysosomes and late endosomes share similar characteristics[3,4], we will refer to them collectively as lysosomes.

There is heterogeneity in lysosomal distribution that is related to functional versatility[2,5]. Lysosomes are distributed between a peripheral pool and a juxtanuclear pool, with dynamic and regulated transport between these two compartments, leading to important functional differences[6,7]. Lysosomal distribution changes in response to various stimuli, and these changes can drive physiological responses in cells[6–12]. During nutrient deprivation, lysosomes cluster near the

nucleus, which facilitates their fusion with autophagosomes to degrade and recycle nutrients[7]. This process can help the cell survive a metabolic emergency. Conversely, upon nutrient replenishment, lysosomes migrate towards the cell periphery where they recruit the mTORC1 complex to their membrane, inducing its activation, driving cell growth and proliferation[13,14]. The translocation of lysosomes also brings mTORC1 close to focal adhesions, which act as membrane hubs to regulate growth factor signaling, amino acid intake, and mTORC1 activity[15,16]. The lysosome-positioning machinery is critical for cytotoxic T lymphocytes and natural killer cells to kill infected or tumor cells by releasing lytic granules, which are lysosome-related organelles or secretory lysosomes containing specific cytotoxic components and lysosomal luminal and membrane proteins[17,18]. Oncogenic transformation also alters physical properties of lysosomes and their distribution, shifting them from the juxtanuclear to the peripheral portion of the cell. This phenomenon is influenced by the tumor

[1]Department of Neurology and Neurosurgery, Montreal Neurological Institute (the Neuro), McGill University, Montreal, QC, Canada. [2]These authors contributed equally: Maleeha Khan, Vincent Francis. ✉e-mail: rahul.kumar@mail.mcgill.ca; peter.mcpherson@mcgill.ca

microenvironment and by gene expression such as elevated levels of kinesin 5B mRNA in cancer tissues and increased association of kinesin 5B with lysosomes. In addition, alterations in proteins that regulate lysosome-positioning can cause psychiatric and neurological disorders. Notably, increased BORCS7 expression links to schizophrenia risk[19,20], mutation in p150-glued subunit of dynactin is associated with amyotrophic lateral sclerosis[21], mutation in kinesin 1B causes Charcot-Marie-Tooth type 2 A[22], and mutation in kinesin 5 C leads to cortical dysplasia with brain malformations type 2[23]. All of these components facilitate the polarized transport of lysosomes in neurons[2,24,25]. Finally, a physiological process that relies entirely on lysosomal positioning is autophagy[26], which participates in precise regulation of the growth, development, and aging of organisms[27]. Autophagy initiates the sequestration of cytoplasmic cargoes or organelles into an autophagosome, which subsequently fuses with a lysosome, forming an autolysosome. Lysosomal hydrolases degrade the materials, releasing breakdown products for future utilization in the cytosol[27].

Small GTPases play a crucial role in regulating lysosomal positioning. These molecular switches alternate between active GTP-bound and inactive GDP-bound states, with GTP hydrolysis to GDP facilitated by GTPase-activating proteins (GAPs) and GDP to GTP conversion mediated by guanine nucleotide exchange factors (GEFs)[28,29]. Small GTPases selectively recruit discrete sets of effector proteins to membranes. The effectors drive the formation of transport vesicles, link organelles to motor proteins, enabling vesicle motility either towards the plus end (anterograde) or the minus end (retrograde) of microtubules[29–32]. Rab GTPases represent the largest branch of small GTPases comprising ~60 members in humans[33,34]. Several members of the Rab GTPase family, including Rab7, Rab26, Rab34, and Rab36, are involved in the regulation of lysosomal positioning. For example, Rab7 binds to Rab interacting lysosomal protein (RILP), leading to the recruitment of cytoplasmic dynein, a minus end directed microtubule motor[35,36]. Rab26, Rab34 and Rab36 can also recruit RILP and promote juxtanuclear clustering of lysosomes[37–41]. In addition, the lysosome-associated tumor suppressor folliculin (FLCN), associated with Birt-Hogg-Dubé (BHD) syndrome, interacts with RILP and Rab34. The formation of FLCN-RILP-Rab34 complex drives the juxtanuclear positioning of lysosomes[42].

In contrast, the anterograde trafficking of lysosomes towards the cell periphery is mediated by kinesin. Arl8a and Arl8b members of the ADP-ribosylation factor (Arf) family of GTPases[43,44], regulate anterograde transport of lysosomes[2]. The recruitment of Arl8 to lysosomes is controlled by BLOC-1-related complex (BORC)[45–47]. Arl8 interacts with SKIP (also known as PLEKHM2) to recruit kinesin-1 for anterograde lysosomal transport[25,45,48–51]. Kinesin-1 requires SKIP for interaction with Arl8, while kinesin-3 interacts directly with Arl8, promoting anterograde transport of lysosomes[52,53]. Thus, Arl8 mediated lysosomal trafficking plays a vital role in a diverse range of essential physiological responses[7,9,13,15,25,45,54–59]. Additionally, there are effectors that alter the direction of GTPase-mediated trafficking. For example, PLEKHM1, a Rab7 effector, competes with SKIP for Arl8b binding and promotes the repositioning of lysosomes[60]. Rab7 and its effector, FYVE and coiled-coil domain-containing protein 1, can recruit kinesin-1 to promote the anterograde movement of lysosomes[61,62]. And, RUFY3 has recently been identified as an effector of Arl8 that links lysosomes to the dynein complex for retrograde transport[63,64].

The DENN (differentially expressed in normal and neoplastic cells) domain bearing proteins constitute the largest family of Rab GEFs[65,66], with poorly known targets and cellular function. Combining proteomics data from human cell map[67] and BioID interactome of Arl8b[63], we report a novel effector of Arl8b, DENND6A, a member of DENN domain protein family. A cell-based screening revealed that DENND6A activates Rab34, allowing recruitment of RILP/dynein to lysosomes. The Arl8b-mediated recruitment of DENND6A to peripheral lysosomes, followed by RILP/dynein recruitment through Rab34 activation, plays a critical role in nutrient-dependent lysosomal positioning

and maintaining steady-state autophagic flux. Our work uncovers a distinct molecular cascade driving the retrograde pathway mediated by Arl8b, which controls autophagy.

## Results

### DENND6A partially localizes to lysosomes

The DENN domain-bearing protein family is composed of 18 proteins grouped into 8 families. To comprehensively unravel their roles in membrane trafficking, we have initiated a long term plan to investigate the specific roles of individual members. To this end, we conducted an extensive search of BioID-based human cell map data[67], which defines the intracellular locations of over 4000 unique proteins, using two independent quantitative approaches. The interactome analysis revealed that one DENN domain protein, DENND6A potentially interacts with LAMP1, LAMP2, LAMP3, LAMTOR1 and STX7 (Fig. 1a), leading to the high confidence prediction that DENND6A localizes to lysosomes. Another BioID interactome identified DENND6A as a potential binding partner of Arl8b[63], a GTPase that primarily localizes to lysosomes and regulates their positioning[2,7].

To investigate if DENND6A localizes to lysosome, we transfected HeLa cells with GFP, or DENND6A-GFP or GFP-DENND6A and stained them with the lysosomal membrane protein LAMP1 (Fig. 1b and Supplementary Fig. 1a). The level of DENND6A overexpression was approximately 1.8 times that of endogenous levels (Supplementary Fig. 2) Both GFP-DENND6A and DENND6A GFP exhibited punctate and tubular structures, showing a partial co-localization with LAMP1 at the juxtanuclear site (Fig. 1b-c and Supplementary Fig. 1a). To validate DENND6A's partial localization to lysosomes, we quantified colocalization using cropped juxtanuclear square-shaped insets from the same images, as represented in Fig. 1b. We compared colocalization before and after a 90-degree rotation of only one channel (LAMP1), a condition that results in random colocalization[68]. We found a significant decrease in colocalization between DENND6A and LAMP1 after the 90-degree rotation (Supplementary Fig. 1c), providing evidence for DENND6A's presence on lysosomes. We also investigated the punctate/tubular DENND6A structures for potential colocalization with early endosomes (EEA1) but did not observe any overlap (Fig. 1c and Supplementary Fig. 1b). DENND6A has also been localized to recycling endosomes[69].

To better visualize the juxtanuclear DENND6A/LAMP1 structures, we performed three-dimensional structured illumination microscopy (SIM) (Fig. 1d), a microscopy technique that provides twice the spatial resolution of confocal microscope, enhancing resolution in both the axial and lateral dimensions[70]. The SIM images indicate that the punctate and tubular structures of DENND6A are associated with LAMP1 vesicles (Fig. 1d). 3D reconstructions revealed that DENND6A was wrapped around the LAMP1 vesicles (Fig. 1e; movie 1; DENND6A in blue and LAMP1 in red). On the other hand, the 3D reconstruction of the SIM image of GFP/LAMP1 did not show any overlap at the juxtanuclear site, with the LAMP1 vesicles situated outside of the GFP region, and most of the bright GFP signal coming from the nucleus (Fig. 1e; movie 2; GFP in blue and LAMP1 in red). In addition, airyscan live-cell microscopy also revealed lysosomal localization of DENND6A (Supplementary Fig. 3a-b, movie 3-4).

Finally, to further test the lysosomal localization of DENND6A, we performed lysosomal immunoprecipitation using hemagglutinin (HA) magnetic beads to immunopurify lysosomes from HEK-293T cells expressing the lysosomal bait protein Tmem192-3xHA, as previously described[71]. As a control, we introduced HEK-293T cells expressing lysosomal bait protein Tmem192-2xFLAG. As expected, we observed the selective enrichment of LAMP1 in the lysosomal fraction, with no other membrane-bound compartments present (Fig. 1f). A pool of DENND6A was also detected in the lysosomal fraction (Fig. 1f). Together, these findings suggest that a pool of DENND6A is localized to the lysosomal compartment at the juxtanuclear site, highlighting the potential functional significance of this localization in lysosomal trafficking.

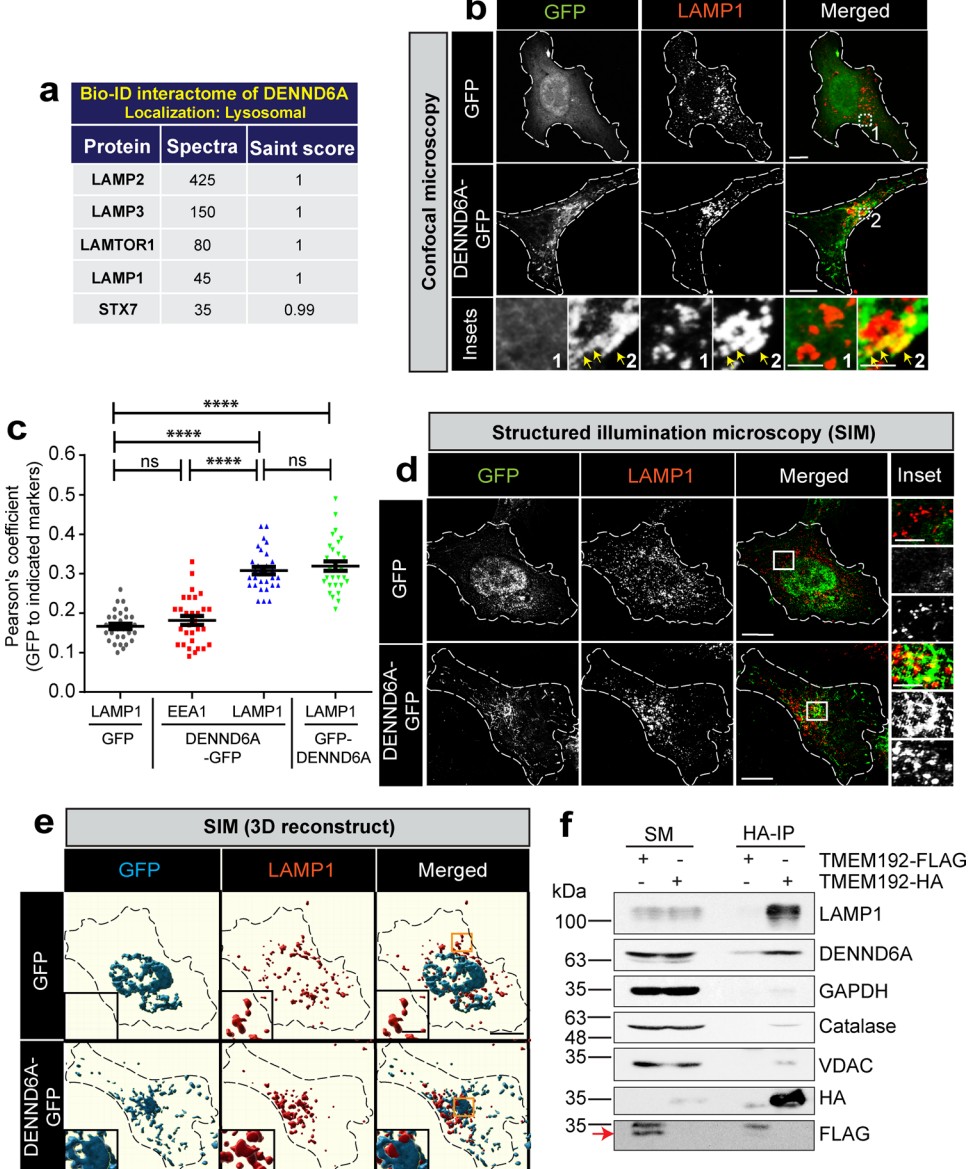

**Fig. 1 | DENND6A-GFP localizes to lysosomes. a** BioID interactome of DENND6A, taken from human cell map. **b** HeLa cells expressing either GFP alone or DENND6A-GFP were fixed, stained with LAMP1 antibody and imaged using confocal microscopy (Leica SP8). The cell periphery is outlined by a white dotted line. Scale bar = 10 μm for low magnification images; 2.8 μm and 2.2 μm for high magnification images corresponding to inset 1 and 2. The yellow arrow indicates colocalization between DENND6A-GFP and LAMP1. **c** Quantification of the Pearson correlation coefficient for the co-localization of GFP with indicated markers (LAMP1, EEA1); means ± SEM; One-way ANOVA (**** P ≤ 0.0001; ns = not significant; *n* = 30 cells for all conditions from 3 replicates). **d** HeLa cells expressing either GFP alone or DENND6A-GFP were fixed, stained with LAMP1 antibody. 3D-SIM images were acquired using LSM880-Elyra PS1 super-resolution microscopy. The cell periphery is outlined by a white dotted line. Scale bar = 10 μm for low magnification images; 4.18 μm and 4.14 μm for high magnification images corresponding to insets from GFP or DENND6A-GFP expressing cells. **e** 3D reconstruction of SIM imaging performed in (**d**). Scale bar = 10 μm for low magnification images; 3 μm for high magnification images. **f** Lysates from HEK-293 cells expressing the Tmem192-3xHA (HA-Lyso cells) or the Tmem192-2xFlag (Control-Lyso cells) were prepared as per the protocol. Lysosomes were immunoprecipitated using anti-HA magnetic beads and analyzed by immunoblot. SM stands for starting material and IP stands for immunoprecipitation. Red arrow indicates specific band corresponding to TMEM192-FLAG.

## DENND6A regulates lysosomal distribution

Arl8b is a critical regulator of lysosomal distribution[2,7,50,51]. Given that DENND6A is a potential binding partner of Arl8b[63], we set out to investigate whether DENND6A also plays a role in the regulation of lysosomal distribution. DENND6A-GFP or GFP-DENND6A expression induced juxtanuclear clustering of LAMP1 (Fig. 2a-c and Supplementary Fig. 1a). To quantitatively assess lysosomal distribution, we employed a well-established and robust unbiased assay[42]. By gradually reducing the perimeter of cell images in 10% decrements (Fig. 2a), we segmented the cells and plotted the cumulative integrated intensity of LAMP1 relative to the entire cell. Statistical comparisons were performed using nonlinear regression and the extra sum of F-squares test. The resulting curves from this analysis should originate from (0,0) and terminate at (100, 1). A leftward shift of the curve indicates a more centralized (juxtanuclear) distribution, while a rightward shift indicates dispersion (Fig. 2a).

Conversely, we investigated the effects of loss of DENND6A on lysosomal distribution. Using CRISPR-Cas9, we generated two independent knockout (KO) lines of DENND6A in HeLa cells, resulting in the loss of DENND6A protein (Fig. 2d–e). Notably, the loss of DENND6A

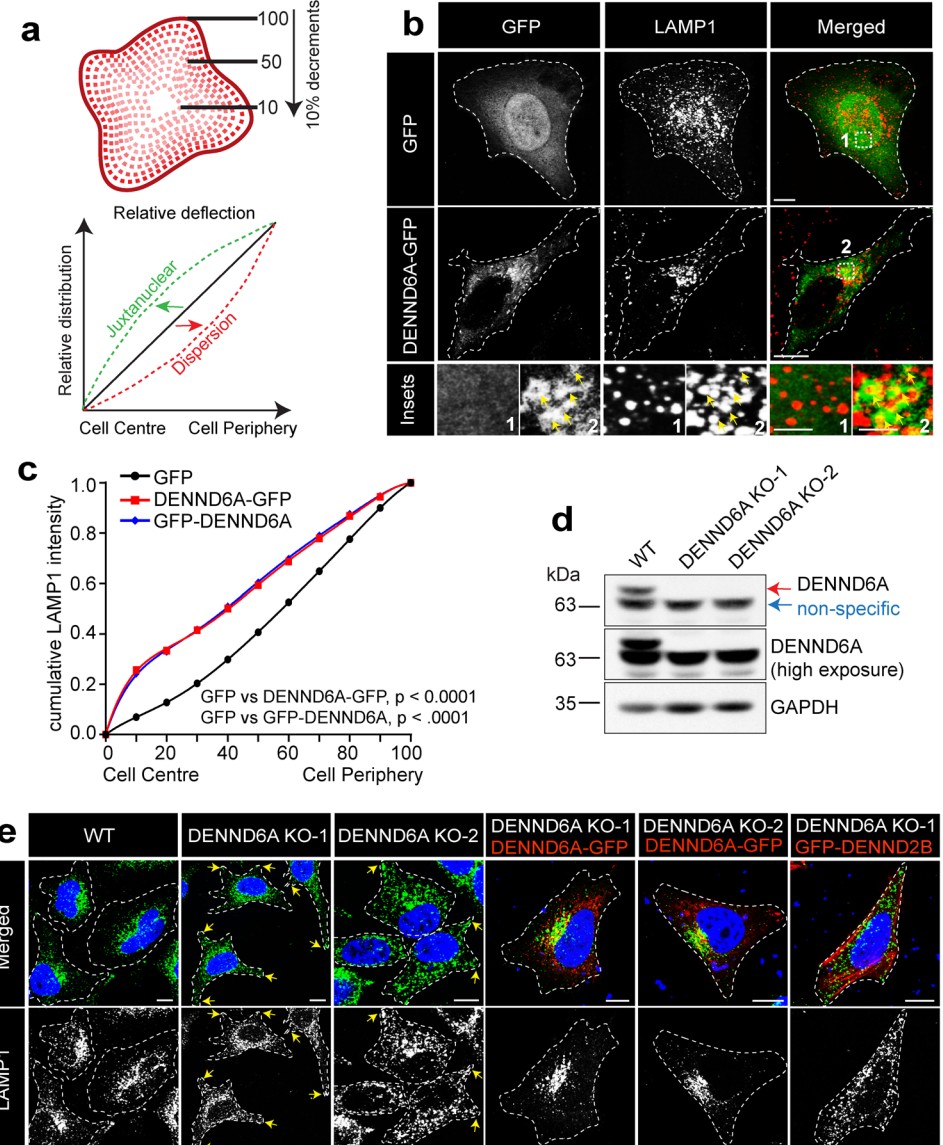

**Fig. 2 | DENND6A promotes juxtanuclear clustering of lysosomes. a** Schematic representation of the quantification of cumulative intensity distribution method for lysosomal distribution (adapted from[42]). **b** Unstarved HeLa cells expressing either GFP alone or DENND6A-GFP were fixed, stained with LAMP1 antibody and imaged using confocal microscopy (Leica SP8). The cell periphery is outlined by a white dotted line. Scale bar = 10 μm for low magnification images; 3.4 μm and 2.1 μm for high magnification images corresponding to inset 1 and 2. **c** Quantification of cumulative distribution of LAMP1 intensity; mean ± SEM; two-tailed extra sum of

F-squares test following nonlinear regression and curve fitting; *n* = 30 cells from 3 replicates. **d** Immunoblot showing the protein levels in WT and DENND6A KO HeLa cells. Immunoblot probed with anti-DENND6A and anti-GAPDH antibodies. **e** WT and DENND6A KO unstarved HeLa cells were fixed, stained with LAMP1 antibody and DAPI, and imaged using confocal microscopy (Leica SP8). The cell periphery is outlined by a white dotted line. Scale bar = 10 μm. Yellow arrow indicates peripheral lysosomes.

caused the dispersion of LAMP1 vesicles towards the cell periphery in both KO lines (Fig. 2e). Expression of DENND6A-GFP in the DENND6A KO cells successfully rescues the lysosomal distribution phenotype (Fig. 2e and Supplementary Fig. 4) whereas expression of another DENN domain protein, DENND2B, does not (Fig. 2e and Supplementary Fig. 4). Collectively, our findings suggest that DENND6A promotes the juxtanuclear clustering of lysosomes.

**Targeting DENND6A to peroxisomes induces peroxisomal juxtanuclear clustering**
Given that DENND6A can drive lysosomal clustering to the juxtanuclear site, we questioned whether DENND6A can induce relocalization of other organelles. For this assay, we selected peroxisomes because they are distributed throughout the cytosol. The assay

involved co-expressing a PEX3-derived peroxisomal targeting signal fused to FK506-binding protein domain (FKBP) and mCherry, and DENND6A fused to FKBP-rapamycin–binding domain (FRB) and GFP (Fig. 3a). This assay relies on an inducible system wherein the introduction of rapamycin triggers the formation of a heterodimer between the FKBP protein from human FKBP12 and the FRB domain of mTOR. As a control, we used a FRB-GFP construct (Fig. 3a). Addition of rapamycin caused the FRB and FKBP domains to interact, directing DENND6A or GFP to the peroxisomes. We observed that in the absence of rapamycin, despite co-expression with DENND6A-FRB-GFP or FRB-GFP, peroxisomes labeled with PEX3-FKBP-mCherry remained scattered throughout the cytoplasm (Fig. 3b-e). However, upon addition of rapamycin, PEX3-FKBP-mCherry labeled peroxisomes clustered at the juxtanuclear site in cells co-expressing DENND6A-FRB-GFP, but not in

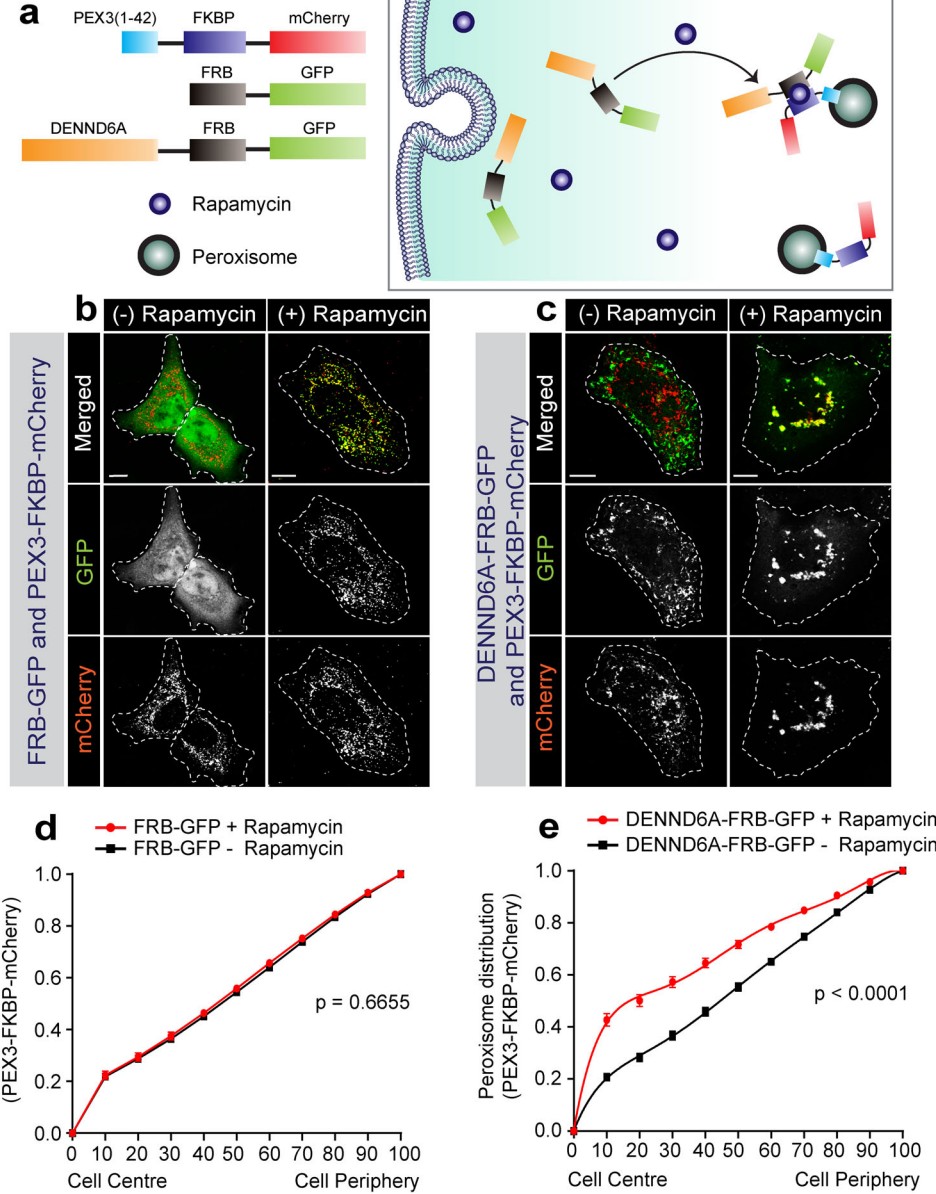

**Fig. 3 | DENND6A is sufficient to promote juxtanuclear clustering. a** Schematic representation of rapamycin-induced relocalization of DENND6A to the peroxisome. FKBP = FK506-binding protein domain; FRB = FKBP-rapamycin–binding domain (adapted from[72]). HeLa cells were co-transfected with (**b**) FRB-GFP and PEX3-FKBP-mCherry, **c** DENND6A-FRB-GFP and PEX3-FKBP-mCherry and treated with or without rapamycin for 1 h. Following rapamycin treatment, cells were fixed and imaged. The cell periphery is outlined by a white dotted line. Scale bars = 10 µm. **d–e** Quantification of cumulative peroxisomal distribution (mCherry intensity) in experiments performed in (**b**) and (**c**); mean ± SEM; two-tailed extra sum of F-squares test following nonlinear regression and curve fitting; n [(GFP-FRB + Rapamycin), (GFP-FRB - Rapamycin), (DENND6A-GFP-FRB + Rapamycin), (DENND6A-GFP-FRB - Rapamycin)] = (31, 30, 30, 29) cells from 3 replicates.

control cells (Fig. 3b-e). These findings demonstrate that targeting DENND6A to an unrelated organelle is sufficient to promote its juxtanuclear clustering.

**Screening for potential substrates of DENND6A**

We next sought to identify Rab substrates for DENND6A and investigate if they contribute to the regulation of lysosomal distribution. We conducted a comprehensive screening of DENND6A against all 60 Rab GTPases using a cell-based assay that involves DENN domain-mediated recruitment of Rabs to mitochondrial membranes (Fig. 4a)[72,73]. Our assay is based on the finding that GEFs play a primary role in driving the spatial and temporal localization of Rab GTPases[74].

We designed a construct called mito-mScarlet-DENND6A, consisting of an import signal of the yeast mitochondrial outer membrane protein Tom70p linked to mScarlet and full length DENND6A, which allows the DENND6A protein to be embedded in the mitochondrial outer membrane but accessible to cytosolic proteins[75,76]. We cotransfected HeLa cells with mito-mScarlet-DENND6A and individual GFP-Rab constructs (Fig. 4b–e) and compared the mitochondrial localization of the GFP-Rabs in the presence of the full length DENND6A targeted to the mitochondria (Fig. 4b–e) or a control mito-mScarlet (Supplementary Fig. 5). Our screening revealed 20 DENND6A/Rab pairs: Rab3B, Rab3C, Rab3D, Rab4A, Rab4B, Rab8A, Rab8B, Rab11A, Rab11B, Rab14, Rab19, Rab22A, Rab22B, Rab25, Rab27A, Rab30, Rab32, Rab34, Rab39A, and Rab43. While the large number of Rabs translocated to the mitochondria was somewhat surprising, it is now evident that DENN domain proteins possess a broader target spectrum than previously appreciated[72,73], which could explain the relative scarcity of

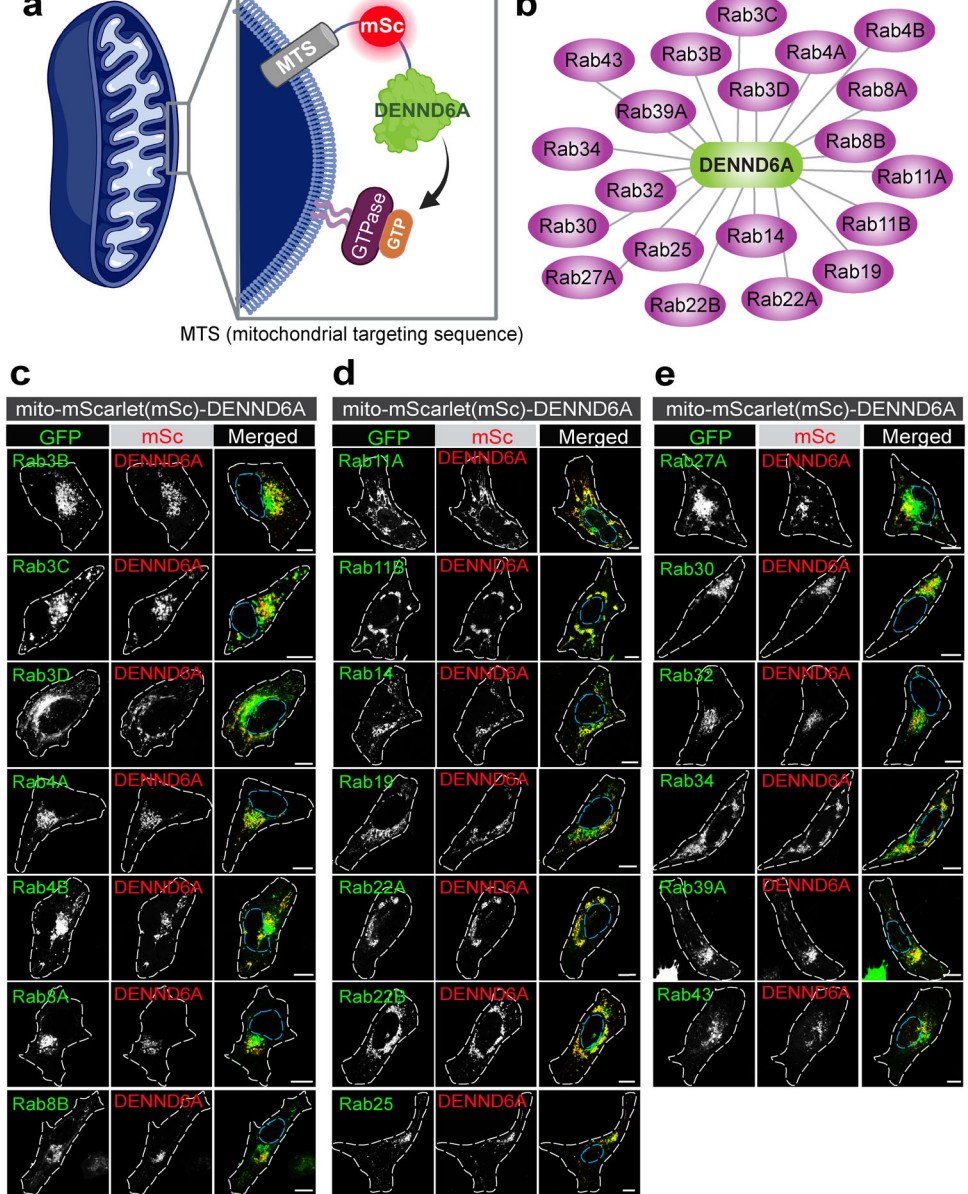

**Fig. 4 | DENN6A targeted to the mitochondria recruits corresponding Rab partners. a** Schematic model of the cell-based assay (adapted from[72]). **b** List of Rabs recruited by DENND6A at the mitochondria. **c–e** HeLa cells co-transfected with GFP-Rabs and mito-mSc-DENND6A were fixed and imaged. The nucleus and cell periphery are outlined by blue and white dotted line respectively. Scale bar = 10 μm.

GEFs compared to the total number of Rab GTPases[28]. Moreover, it is also possible that among the identified pairs, some may serve as GEF substrates while others act as binding partners but not as substrates. Therefore, it will be important to further characterize the pairs identified.

## DENND6A-mediated Rab34 activation recruits RILP/dynein to promote lysosomal juxtanuclear clustering

After identifying new pairs, our focus turned to determining which Rab is involved in the lysosomal clustering phenotype mediated by DENND6A. Among all the Rabs identified as DENND6A interactors, Rab34 emerged as a promising candidate due to its known impact on promoting juxtanuclear clustering of lysosomes[39,40,42,77]. Both the WT or active mutant form of Rab34 (Rab34 Q111L) promote juxtanuclear clustering of lysosomes as compared to the inactive mutant form (Rab34 T66N)[78]. This led us to investigate the role of Rab34 in DENND6A-mediated lysosomal distribution.

We first sought to examine the relationship between DENND6A and Rab34. We performed binding experiments using purified GST-Rab34 QL and GST-Rab34 TN mutants incubated with lysates from cells transfected with DENND6A-GFP. We found that there was a preferential binding to the inactive (TN) form of Rab34 mutant (Fig. 5a, b), a hallmark of GEFs[66,79–81]. We also used an effector binding assay with a T7 tag fused to RILP (a Rab34 effector), which selectively binds to the active form of Rab34[39]. We found that the active levels of Rab34 were increased in cells transfected with DENND6A-GFP compared to control cells expressing GFP (Fig. 5c, d). Finally, to further confirm the GEF activity of DENND6A toward Rab34, we performed an in-vitro GEF assay, which showed that the amount of GTP loaded Rab34 was significantly greater when Rab34 was incubated with DENND6A (Fig. 5e).

Next, we wondered if DENND6A and Rab34 function together in cells promoting juxtanuclear lysosomal clustering. DENND6A-GFP and mCherry-Rab34 both clustered with LAMP1 vesicles in WT HeLa cells (Fig. 5f). To enhance our visualization of this clustering, we utilized SIM

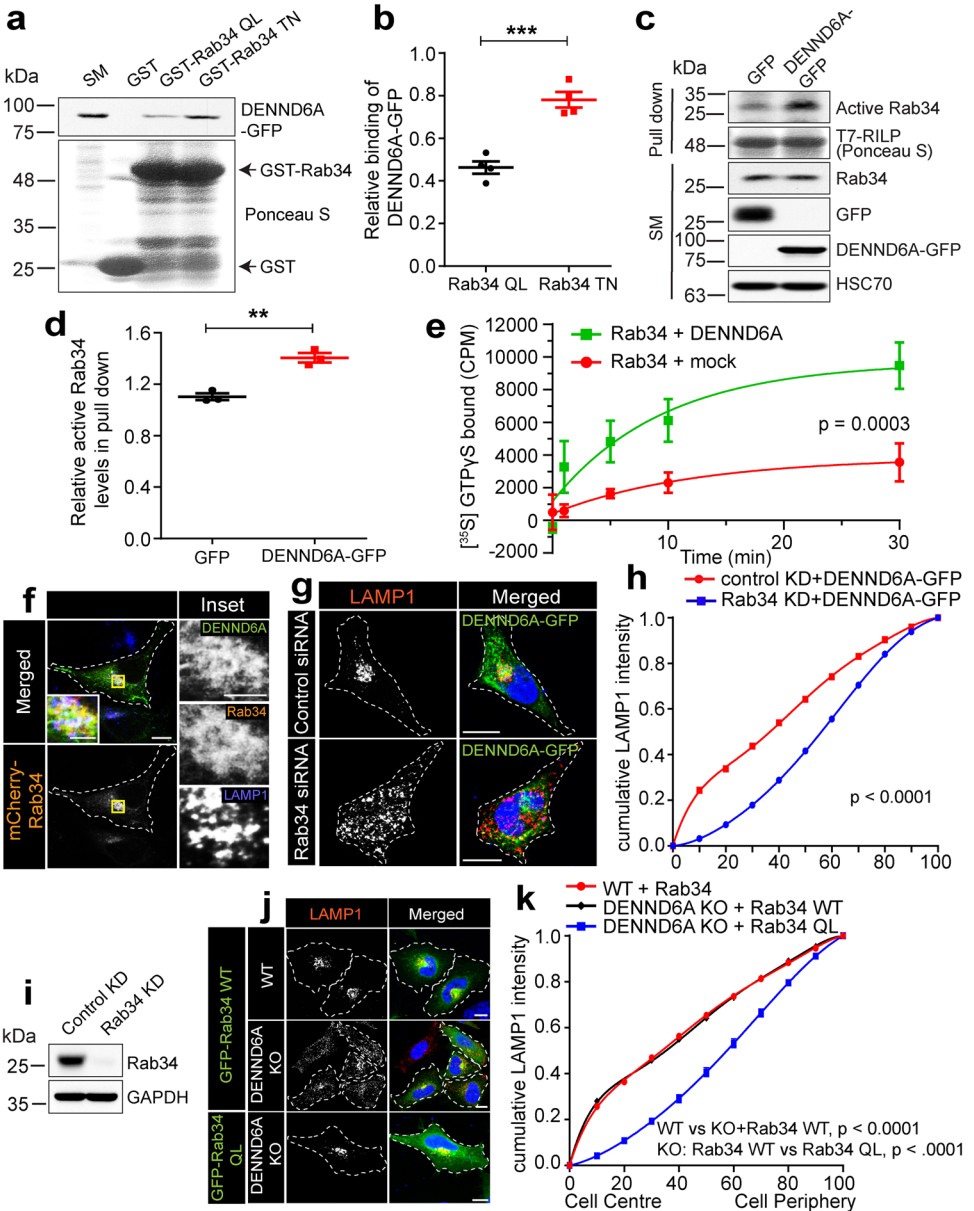

**Fig. 5 | DENND6A promotes juxtanuclear clustering of lysosomes via activation of Rab34. a** HEK-293T cell lysates expressing DENND6A-GFP were incubated with indicated purified proteins. Bound proteins were detected by immunoblot with anti-GFP antibody. Starting material (SM). **b** Quantification from (**a**); means ± SEM; two-tailed unpaired t test (*** P ≤ 0.0005; n = 4 from three replicates). **c** GFP or DENND6A-GFP expressing HEK-293T cell lysates were incubated with purified T7-RILP. Bound proteins were detected by anti-Rab34, anti-GFP and anti-HSC70 antibody. **d** Quantification from **c**; means ± SEM; two-tailed unpaired t test (**P ≤ 0.0025; n = 3 from three replicates). **e** In vitro GEF assays using purified Rab34 with or without DENND6A as indicated. Relative incorporation of [$^{35}$S]GTPγS on Rab34 is plotted over time; data represent mean ± SEM; two-tailed extra sum of F-squares test following nonlinear regression one-phase association curve fit, n = 3 from three replicates. **f** HeLa cells co-expressing DENND6A-GFP and mCherry-Rab34 were fixed/stained with LAMP1 antibody. Cell periphery outlined by a white dotted line. Scale bar = 10 and 3.13 μm for low and high magnification images.

**g** HeLa cells treated with control or Rab34 siRNA were transfected with DENND6A-GFP. Post transfection, cells were fixed and stained with LAMP1 antibody. Cell periphery is outlined by a white dotted line. Scale bar = 10 μm. **h** Quantification of cumulative LAMP1 distribution from **g**; mean ± SEM; two-tailed extra sum of F-squares test following nonlinear regression and curve fitting; n = 30 cells from 3 replicates. **i** Immunoblot showing the Rab34 protein levels in control and Rab34 siRNA treated HeLa cells. Immunoblot probed with anti-Rab34 and anti-GAPDH antibodies. **j** WT and DENND6A KO HeLa cells were transfected with GFP-Rab34 or GFP-Rab34 QL. Post transfection, cells were fixed and stained with LAMP1 antibody. Cell periphery is outlined by a white dotted line. Scale bar = 10 μm. **k** Quantification of cumulative LAMP1 distribution from (**j**); mean ± SEM; two-tailed extra sum of F-squares test following nonlinear regression and curve fitting; n = 27, 30 and 30 cells from 3 replicates corresponding to the three conditions Rab34+WT; Rab34 + KO1; Rab34 QL + KO1.

and observed the overlapping regions between LAMP1 vesicles and both DENND6A and Rab34 (Supplementary Fig. 6a). Fluorescently tagged Rab34 is known to localize to the Golgi[42]. While SIM imaging of DENND6A/Rab34/GM130 revealed a substantial overlap between Rab34 and GM130, DENND6A exhibited minimal overlap with GM130 (Supplementary Fig. 6b), indicating that DENND6A is not localized to

Golgi. To test if DENND6A can promote lysosomal clustering in the absence of Rab34, we performed Rab34 knockdown (KD) experiments. DENND6A-GFP failed to promote juxtanuclear LAMP1 clustering in Rab34 KD cells, as opposed to control KD cells (Fig. 5g–i). Conversely, in DENND6A KO cells, the expression of GFP-Rab34 WT failed to cluster LAMP1 vesicles as observed in WT cells (Fig. 5j–k). We also observed a

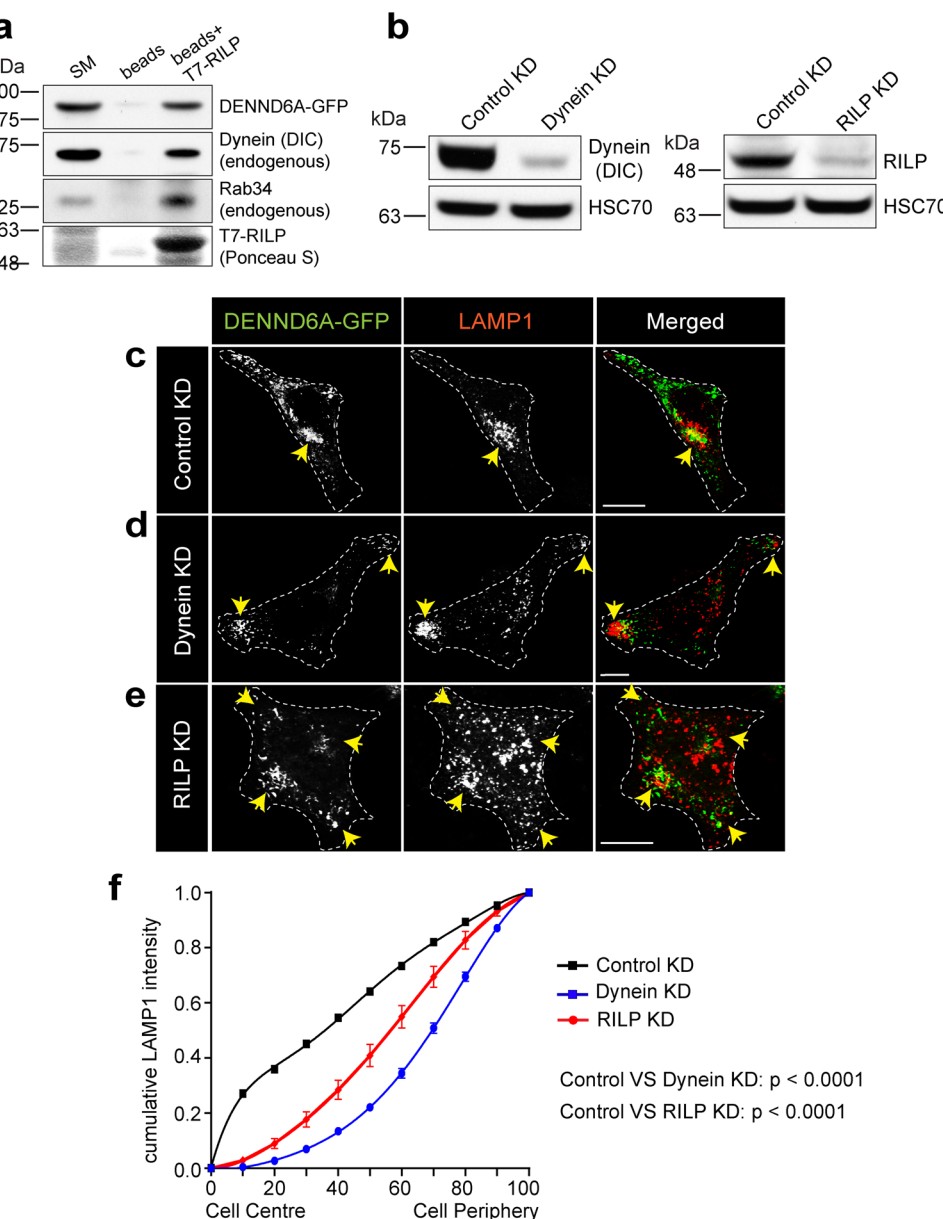

**Fig. 6 | DENND6A mediated juxtanuclear lysosomal clustering depends on RILP and dynein. a** Lysates from HEK-293T cells expressing DENND6A-GFP were incubated with wither beads covalently linked to T7 antibody or T7 linked beads coupled to T7-RILP protein. Specifically bound proteins were detected by immunoblot with anti-GFP antibody, anti- dynein intermediate chain (DIC) antibody and anti-Rab34 antibody. The starting material (SM) was run in parallel to detect the total DENND6A-GFP, DIC and Rab34. **b** Immunoblot showing DIC protein levels in control and dynein siRNA treated HeLa cells. Immunoblot probed with anti-DIC and anti-HSC70 antibodies. Immunoblot showing RILP protein levels in control and RILP siRNA treated HeLa cells. Immunoblot probed with anti-RILP and anti-HSC70 antibodies. HeLa cells were treated with (**c**) control siRNA or (**d**) dynein siRNA or (**e**) RILP siRNA and transfected with DENND6A-GFP. 16 h post transfection, cells were fixed and stained with LAMP1 antibody. The cell periphery is outlined by a white dotted line. The yellow arrow indicates the presence of lysosomes corresponding to DENND6-GFP. Scale bars = 10 μm. **f** Quantification of cumulative distribution of LAMP1 intensity in experiments performed in (**c**)–(**e**); mean ± SEM; two-tailed extra sum of F-squares test following nonlinear regression and curve fitting; $n = 29, 30, 30$ cells from 3 replicates corresponding to control, dynein, and RILP siRNA treated cells.

reduction in the localization of GFP-Rab34 to lysosomes in DENND6A KO cells (Supplementary Fig. 7). Furthermore, the expression of GFP-Rab34 QL (constitutively active mutant) in DENND6A KO cells restores the lysosomal distribution pattern to that observed in WT/GFP-Rab34 cells (Fig. 5j-k). Collectively, these findings indicate that DENND6A and Rab34 function together in promoting lysosomal clustering.

Rab34 mediated juxtanuclear clustering requires interaction with its effector RILP[39,78]. RILP is a proposed dynein adapter which is essential for promoting retrograde transport of lysosomes[82–84]. To explore the molecular mechanism downstream of Rab34, we

performed a pulldown experiment using purified T7-RILP incubated with lysates from cells transfected with DENND6A-GFP. This revealed the presence of a complex containing DENND6A, Rab34, dynein, and RILP (Fig. 6a). We also conducted a co-immunoprecipitation experiment and identified interactions between DENND6A and Rab34, RILP, dynein, and dynactin (p150) (Supplementary Fig. 8).

We next tested the role of RILP and dynein in DENND6A mediated lysosomal clustering. We performed knockdown (KD) of either protein in cells expressing DENND6A-GFP (Fig. 6b). Both dynein and RILP KD prevented clustering of lysosomes at the juxtanuclear site driven by

DENND6A expression. Instead, we observed peripheral (for dynein KD) or dispersed (for RILP KD) distribution (Fig. 6c-f). Interestingly, cells with dynein KD displayed dispersion of nearly all juxtanuclear DENND6A punctate and tubular structures to the periphery where they were largely associated with LAMP1-marked lysosomes (Fig. 6d). Finally, we demonstrate that the ability of DENND6A to induce peroxisome clustering depends on RILP and dynein (Supplementary Fig. 9). This suggests that both dynein and RILP are required downstream of DENND6A/Rab34 to promote juxtanuclear clustering of lysosomes.

## DENND6A binding to Arl8b drives juxtanuclear clustering of lysosomes

A BioID based interactome identified DENND6A as a potential binding partner of Arl8b[63]. To test the relationship between Arl8 and DENND6A, we conducted binding experiments using purified GST-Arl8b Q75L and GST-Arl8b T34N mutants incubated with lysates from cells transfected with DENND6A-GFP. DENND6A bound preferentially to the active (QL) form of Arl8b (Fig. 7a, b). In addition, DENND6A-GFP immunoprecipitates the active form but not the inactive form of Arl8b (Supplementary Fig. 10). As DENND6A prefers interaction with the active form of Arl8b, suggesting that DENND6A is an Arl8b effector, we tested if targeting active Arl8b to mitochondria recruits DENND6A to these organelles. However, our experiments did not indicate any recruitment by either the active or inactive Arl8b mutants (Supplementary Fig. 11). It is possible that the lipid composition of the lysosomal membrane may also influence DENND6A recruitment, a hypothesis that should be explored in future studies. Similar recruitment phenomena have been observed for Arf GTPase effectors like four-phosphate-adapter protein 1 and 2 (FAPP1 and FAPP2), which use their pleckstrin homology domain to interact with PtdIns(4)P and Arf for localization to the trans-Golgi network[85]. Nevertheless, the biochemical evidence from our pulldown and immunoprecipitation experiments aligns with the characteristic binding of an effector protein to a GTPase, which typically occurs preferentially in the GTP-bound state[86,87]. Furthermore, we observed co-localization between Arl8b-mCherry and DENND6A-GFP in cells (Fig. 7c, d).

The presence of a specific effector at the site of GTPase activation determines the route of downstream signaling events[86,87]. Arl8b links lysosomes to kinesin, facilitating anterograde transport and leading to peripheral lysosome accumulation[2,7,50,51]. Confirming these findings, we observed that Arl8b overexpression led to a peripheral distribution of lysosomes (Fig. 7d, e). Intriguingly, when Arl8b-mCherry and DENND6A-GFP were co-expressed, a significant shift was observed in the distribution of both Arl8b and LAMP1-marked lysosomes toward the juxtanuclear site (Fig. 7d, e).

We next investigated the impact of double KD of Arl8a and Arl8b on DENND6A localization. The double KD significantly reduced the percentage of cells exhibiting a juxtanuclear pool of DENND6A, and the juxtanuclear DENND6A punctate and tubular structures were faintly observed (Fig. 7f–h). Finally, we observed that DENND6A localization to lysosomes was reduced in Arl8 KD cells (Fig. 7f and Supplementary Fig. 12). Thus, our findings suggest that DENND6A functions as an effector of Arl8b, facilitating the juxtanuclear positioning of lysosomes.

## DENND6A regulates nutrient dependent lysosomal positioning and autophagic flux

Our findings suggest that Arl8b facilitates the recruitment of DENND6A to lysosomes, leading to activation of Rab34. Rab34, in turn, recruits its effector RILP, a dynein adapter that promotes retrograde transport of lysosomes. Previous studies have demonstrated that Arl8 and its upstream regulator BORC control the positioning of lysosomes in response to nutrient levels, and nutrient starvation causes juxtanuclear clustering of lysosomes[7,45,46]. We observed that DENND6A KO resulted in an increased distribution of lysosomes towards the cell

periphery under normal conditions (Fig. 8a). Given that nutrient starvation induces lysosomal clustering, we wondered if this phenomenon would occur in DENND6A KO cells. Indeed, we found that nutrient starvation caused the juxtanuclear clustering of LAMP1-marked lysosomes in WT cells (Fig. 8a, b), whereas lysosomes in DENND6A KO cells remained significantly peripheral even under starvation conditions (Fig. 8a, b).

The clustering of lysosomes in response to decreased nutrient levels is known to promote the fusion between autophagosomes and lysosomes[88]. This fusion is crucial for coordinating autophagic flux[7], which is measured by the amount of lipidated, autophagosome-associated form of LC3 (LC3-II) remaining in cells[89]. We assessed the levels of LC3B-II in unstarved WT and DENND6A KO cells. We observed significantly lower levels of LC3B-II in DENND6A KO cells (Fig. 8c, d). The decrease in LC3B-II levels could result from two scenarios: reduced autophagosome formation, leading to decreased autophagy, or increased fusion between autophagosomes and lysosomes, resulting in enhanced autophagic degradation[7,90]. When we induced autophagy through nutrient starvation, which increases autophagosome synthesis to provide necessary nutrients[7,91], we still observed significantly reduced levels of LC3B-II in DENND6A KO cells (Fig. 8c, d). To test whether the decrease in LC3B-II levels resulted from increased autophagosome-lysosome fusion, we treated the cells with Bafilomycin A1 (BafA1), an inhibitor of lysosomal acidification that blocks autophagosome-lysosome fusion and degradation. Despite the inhibition of fusion, we still observed lower levels of LC3B-II (Fig. 8c, d), suggesting that DENND6A primarily affects autophagosome formation rather than autophagosome-lysosome fusion. We also confirmed the lower levels of LC3B-II in DENND6A KO cells using immunofluorescence (Fig. 8e, f), suggesting impaired autophagic flux in the absence of DENND6A. The distribution of LC3B also showed a modest dispersion in the DENND6A KO cells when compared to the WT (Supplementary Fig. 13), although not as extensive as we observed for lysosomes. In addition, we also assessed the levels of the autophagy cargo receptor p62 (Fig. 8c and Supplementary Fig. 14). Under unstarved conditions, DENND6A KO cells had reduced p62 levels. This reduction could stem from either decreased autophagy or increased autophagic degradation. However, if the latter were the case, treatment with BafA1/starvation (which blocks autophagosome-lysosome fusion and degradation) should have stabilized p62 levels to match those of the WT, but this did not occur. These observations suggest that the loss of DENND6A leads to reduced autophagy.

Lysosomal positioning plays a crucial role in coordinating mTORC1 activity, thereby regulating autophagosome synthesis and autophagosome-lysosome fusion[7]. Specifically, mTORC1 phosphorylates ULK1 on Ser 758 to inhibit ULK1's interaction with and activation by AMPK, subsequently preventing autophagy[92]. However, during starvation, mTORC1 is inhibited, ULK1 Ser758 phosphorylation is reduced, allowing ULK1 to interact with AMPK and initiate autophagy[91,92]. Given reduced autophagy, we investigated if autophagy initiation is blocked by probing for ULK1 Ser758 phosphorylation (Supplementary Fig. 15). Our findings indicate that while ULK1 undergoes dephosphorylation under nutrient starvation in WT cells, the levels of phospho-ULK1 (Ser758) were significantly higher in DENND6A KO cells during starvation. This suggests that autophagy initiation is also impaired in the absence of DENND6A.

In addition to its role in autophagic degradation, lysosomes are involved in regulating various other cellular processes, such as endocytic and phagocytic degradation[2]. To further investigate DENND6A's impact, we examined its influence on endocytic cargo delivery to lysosomes by subjecting both WT and DENND6A KO cells to DQ-BSA. DQ-BSA is an endocytic cargo that fluoresces upon proteolytic cleavage within lysosomes[60,93,94]. Our observations revealed a significant reduction in the fluorescent intensity of internalized cargo (DQ-BSA) in DENND6A KO cells compared to WT cells (Supplementary Fig. 16a-b).

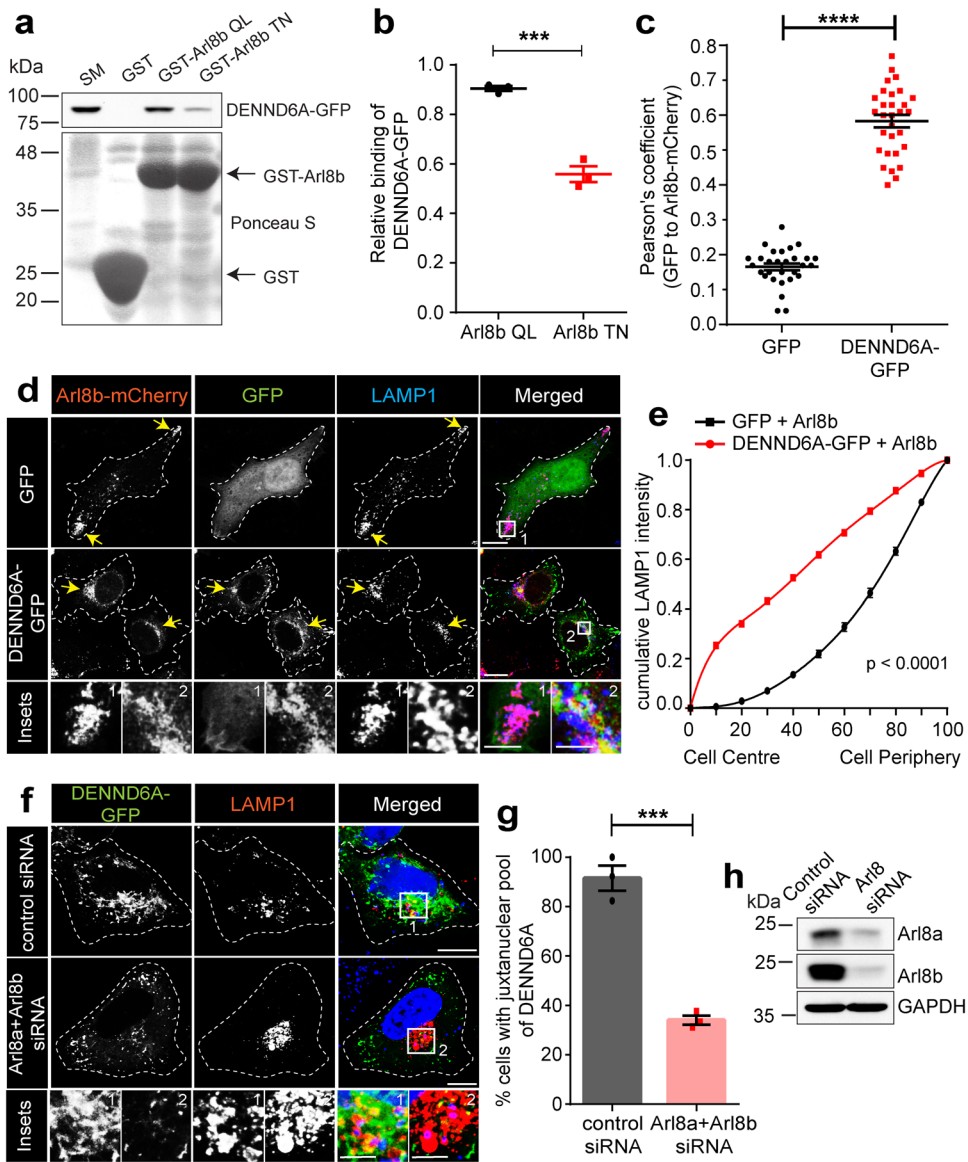

**Fig. 7 | DENND6A is an Arl8b effector, and their interaction promotes juxta-nuclear lysosomal clustering. a** Purified GST, GST-Arl8b QL and GST-Arl8b TN were incubated with lysates from HEK-293T cells expressing DENND6A-GFP. Bound proteins were detected by immunoblot with anti-GFP antibody. The starting material (SM) was run in parallel to detect the total DENND6A-GFP. **b** Quantification of experiment in (**a**); means ± SEM; two-tailed unpaired t test (*** P ≤ 0.0005; $n = 3$ from three replicates). **c** Graphical representation of the Pearson correlation coefficient for the co-localization of GFP or DENND6A-GFP with Arl8b-mCherry from experiments performed in (**d**); means ± SEM; two-tailed unpaired t test (****$P ≤ 0.0001$; $n = 29$ and 30 corresponding to GFP and DENND6A-GFP, from 3 replicates). **d** HeLa cells co-expressing either GFP alone or DENND6A-GFP along with Arl8b-mCherry were fixed/stained with LAMP1 antibody and imaged (confocal microscopy). Cell periphery is outlined by a white dotted line. The yellow arrow indicates the presence of lysosomes corresponding to Arl8b-cherry in the presence of either GFP or DENND6-GFP. Scale bar = 10 μm for low magnification images; 5.22 μm and 2.12 μm for high magnification images corresponding to inset 1 and 2. **e** Quantification of cumulative distribution of LAMP1 intensity in (**d**); mean ± SEM; two-tailed extra sum of F-squares test following nonlinear regression and curve fitting; $n = 25$ and 26 cells, corresponding to GFP and DENND6A-GFP, from 3 replicates. **f** Control or Arl8 siRNA treated HeLa cells were transfected with DENND6A-GFP. 16 h Post transfection, cells were fixed/stained with LAMP1 anti-body and imaged (confocal microscopy). The cell periphery is outlined by a white dotted line. Scale bar = 10 μm for low magnification images; 3.79 μm and 4.97 μm for high magnification images corresponding to inset 1 and 2. **g** Quantification of experiment in (**f**); means ± SEM; two-tailed unpaired t test (***$P = 0.0004$; $n = 42$ and 47, corresponding to control and Arl8 siRNA, from 3 replicates). **h** Immunoblot showing Arl8a and Arl8b protein levels in control and Arl8(a + b) siRNA treated HeLa cells. Immunoblot probed with anti-Arl8a, anti-Arl8b and anti-GAPDH antibodies.

This suggests that the absence of DENND6A impairs endocytic cargo degradation.

## Discussion

The dynamic regulation of lysosomal positioning is vital for lysosomal function. Organelles and vesicles can undergo movement between different areas of the cell along microtubules. Anterograde transport is facilitated by kinesin motors, while retrograde transport is driven by cytoplasmic dynein. The coupling of lysosomes to molecular motors governs their distribution and a repertoire of cellular physiology[2,5]. Interactions between organelles can also influence positioning, for example the endoplasmic reticulum (ER) plays a role in shaping endosomal distribution for regulated cargo transportation[62,95–97]. Moreover, the positioning of lysosomes actively influences remodeling of organelles such as the ER in the outer periphery[98,99]. A wide range of machineries, including GTPases,

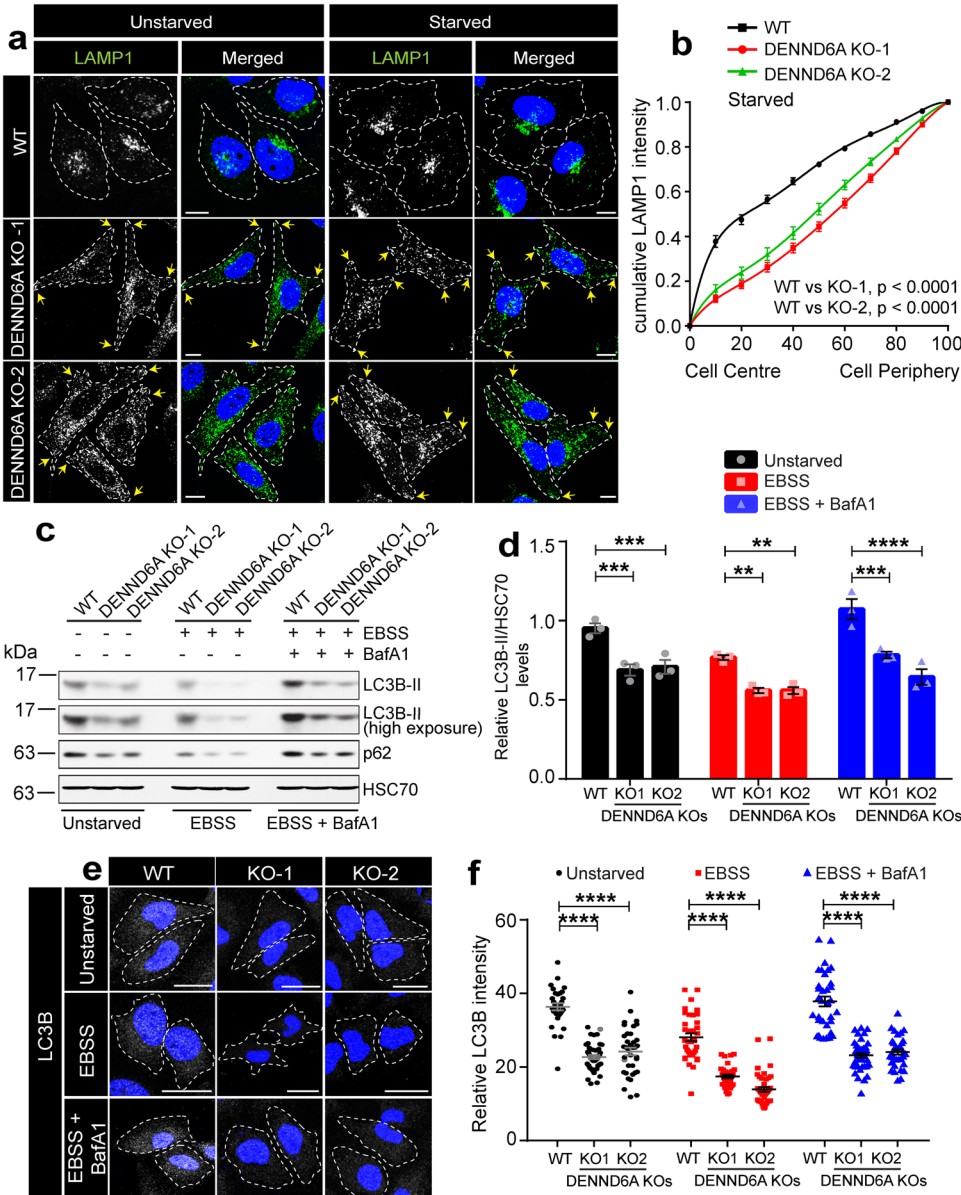

**Fig. 8 | Loss of DENND6A impairs nutrient dependent lysosomal positioning and autophagic flux. a** Unstarved or Earle's Balanced Salt Solution (EBSS) starved HeLa cells were fixed, stained with LAMP1 antibody and imaged using confocal microscopy (Leica SP8). The cell periphery is outlined by a white dotted line. Scale bar = 10 μm. Yellow arrows indicate the presence of peripheral lysosomes. **b** Quantification of cumulative distribution of LAMP1 intensity in experiments performed in (**a**) (under starvation condition); mean ± SEM; two-tailed extra sum of F-squares test following nonlinear regression and curve fitting; n = 28, 29, 30 cells, corresponding to WT, DENND6A KO1 and DENND6A KO2, from 3 replicates. **c** Immunoblot showing LC3B-II protein levels under various conditions (unstarved; EBSS starved; and EBSS starved + Bafilomycin A1 (BafA1)) Immunoblot probed with anti-LC3B-II, anti-p62 and anti-HSC70 antibodies. **d** Quantification of experiment in (**c**); means ± SEM; two-way ANOVA (**P ≤ 0.0025; ***P ≤ 0.0005; ****P ≤ 0.0001; n = 3 from three replicates). **e** HeLa WT and DENND6A KOs cells from **c** were fixed and stained with LC3B-II antibody and DAPI. The cell periphery is outlined by a white dotted line. Scale bar = 25 μm. **f** Quantification of experiment in (**e**); means ± SEM; Kruskal-Wallis test ****P ≤ 0.0001; n = (32, 43, 37); (36, 34, 42); (34, 41, 34) cells, corresponding to WT, DENND6A KO1 and DENND6A KO2 in unstarved; EBSS; EBSS + BafA1, from 3 replicates).

effector proteins, and adapter proteins, have been identified as coupling factors for motor proteins[1]. A major regulatory GTPase on lysosomes is Arl8b. Here, we report the identification of DENND6A as an Arl8b effector that couples lysosomes to RILP-dynein-dynactin via activation of Rab34, promoting retrograde transport of lysosomes (Fig. 9).

In a previous study, DENND6A was found to function as a GEF for Rab14. DENND6A/Rab14 interaction defines an endocytic recycling pathway that regulates cell-cell junctions[69]. Interestingly, Rab14 was also identified in our screen, underscoring the reliability of our cell-based assay approach. We discovered 20 hits associated with

DENND6A screening. While we have established the GEF/substrate relationship between DENND6A and Rab34, further investigations are necessary to unravel the relationships between other pairs and their specific cellular functions. It is not surprising that we associated only one pair, DENND6A/Rab34, with lysosomal distribution, as only a small fraction of DENND6A is linked to lysosomes. We still need to explore and characterize the majority fraction of DENND6A, which is likely to involve other Rab hits. Given the broad spectrum of potential targets for DENND6A, it also remains plausible that interactions with other compartments under different physiological conditions, each harboring DENND6A and other Rab partners, may exist. These

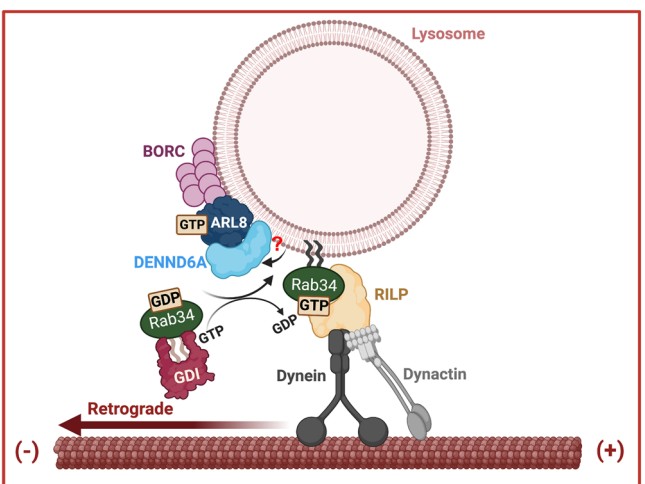

**Fig. 9 | Schematic representation of Arl8b/DENND6A/Rab34 mediated retrograde lysosomal trafficking.** We propose that DENND6A functions as an effector of Arl8b. Furthermore, we hypothesize the presence of an additional factor (marked by '?') that may assist in DENND6A's recruitment to lysosomes. The BORC complex recruits Arl8b to peripheral lysosomes (adapted from[63]), where Arl8b in turn recruits DENND6A. Once recruited, DENND6A activates Rab34, and the activated Rab34 then recruits its own effector, RILP, which is a proposed dynein adapter. Consequently, DENND6A links Arl8b and Rab34, facilitating retrograde lysosomal transport, from the plus to minus ends of the microtubules.

interactions are likely to become more apparent as future studies on these other pairs are reported.

In addition to their involvement in autophagy, lysosomes play a crucial role in regulating various cellular functions such as cell adhesion, invasion, programmed cell death, and metabolic signaling[2]. It is plausible that the regulation of cell-cell junctions mediated by DENND6A and Rab14 could be interconnected through a series of signaling events within the lysosomal system. However, it is worth considering that DENND6A-mediated regulations through other Rabs may not necessarily involve only lysosomes. Therefore, alternative mechanisms or pathways may be involved in the DENND6A-mediated regulations of other Rabs. Furthermore, the KD of Rab25, one of the identified hits among the 20, resulted in the loss of β1-integrin and cell migration defects[100]. Interestingly, a similar phenotype was observed upon loss of DENND6A[69], suggesting a potential importance of the DENND6A/Rab25 pair that warrants further investigation.

Our study unveils a unique role of DENND6A as an effector of Arl8b, facilitating its crucial interaction with peripheral lysosomes. While biochemical evidence and the relocalization of peripheral lysosomes to the juxtanuclear site upon coexpression of DENND6A/Arl8b support DENND6A's role as an effector, it's worth noting that there may be an additional, as yet unknown factor facilitating DENND6A's recruitment to lysosomes (Fig. 9). This is suggested by the failure of Arl8b alone at the mitochondria to recruit DENND6A. In addition, we have demonstrated that DENND6A acts as a critical link, coupling lysosomes to RILP-dynein-dynactin for their retrograde transport towards the juxtanuclear site through the activation of Rab34. While fluorescently tagged Rab34 has been reported to localize to the Golgi, there may be potential crosstalk between lysosomes at the juxtanuclear site and the Golgi during this process. Typically, inactive Rabs are cytosolic and bound to GDP-dissociation inhibitors[74,101]. They can be recruited directly to their respective membrane organelles once their GEF is localized to the membrane[74,101]. Given these findings and the fact that the expression of Rab34 leads to complete lysosomal clustering, we propose that DENND6A is recruited by Arl8b to peripheral lysosomes (Fig. 9). Subsequently, the recruitment of DENND6A

can lead to the recruitment of inactive Rab34 from the cytosol to the lysosomes, following the established Rab biology principles.

The Arl8b/DENND6A/Rab34/RILP interactions play a crucial physiological role in controlling autophagic flux. Notably, the loss of DENND6A leads to a reduction in autophagy, potentially through impairment of autophagosome formation. Interestingly, the 'relative flux' appears similar across conditions (WT versus KO1 versus KO2), indicating that EBSS reduces LC3B-II levels, while EBSS+BafA1 treatment increases LC3B-II levels. Although LC3B-II levels appear lower in the KO cells, this reduction cannot be solely attributed to the autophagy block but may also be influenced by lower expression levels, as evidenced by the levels of p62. Could this also be associated with mTOR dysregulation? Given that mTOR is activated at lysosomes, our observation of impaired starvation-induced dephosphorylation of mTOR-dependent ULK1 phosphorylation raises questions about the dephosphorylation status of other mTOR substrates, including TFEB. If TFEB remains phosphorylated, it may be unable to translocate to the nucleus, hindering the activation of transcription for lysosome/autophagy genes[102,103]. This could explain the observed lower levels of LC3B-II and p62. Further studies are needed to elucidate the mechanism by which the loss of DENND6A impairs autophagy and the impact of DENND6A on autophagosome formation. Furthermore, our findings indicate that the absence of DENND6A also hinders endocytic cargo degradation, a phenotype that necessitates additional experiments in the future to pinpoint the exact dysfunction within this pathway.

While the role of Arl8b in promoting anterograde transport of lysosomes via its binding to the effector protein SKIP, which recruits the kinesin-1 motor, has been previously recognized, recent studies have revealed that Arl8b can also recruit RUFY3 to promote retrograde transport[63,64]. RUFY3 can directly bind to dynein[63] or indirectly associate with dynein through the dynein activating adapter JIP4[64]. However, unlike DENND6A-mediated retrograde transport, RUFY3-mediated retrograde transport of lysosomes does not impact autophagy[64]. Given the identification of multiple effectors/adapters for Arl8b, it is probable that these effectors compete for binding to Arl8b. The probability of their binding might also depend on the levels of Arl8b relative to these effectors, which may vary depending on cell/tissue type. It raises the question of whether there is sufficient Arl8b to bind to all effectors simultaneously? Exploring these possibilities and understanding the specific conditions under which these molecules are recruited will be a significant focus for future research.

Arl8 has also been associated with exosome secretion[104]. Exosomes are small vesicles that originate from various intracellular compartments such as endosomes, lysosomes, autophagosomes and other compartments[105]. Exosomes mediate crucial role in intercellular communication and have been known to play important function in immune response, cancer, development and neurodegenerative disorders[106]. Loss of Arl8 led to an increase in exosome secretion[104]. It would be interesting to explore in the future if DENND6A influences exosome release.

Our findings contribute to the growing complexity of signaling pathways by identifying a new aspect of regulation in the Arl8b-mediated retrograde transport system. We have discovered that DENND6A serves as an effector of Arl8b, shedding light on yet another mechanism involved in this process. Notably, through downstream of binding to Arl8b, DENND6A activates Rab34, which in turn engages its effector, RILP, to recruit dynein. This finding is intriguing because RILP is also a shared effector for Rab7[35]. Additionally, several other proteins have been previously identified, such as TMEM55B[107], JIP3/JIP4[107,108], SEPT9[109], and TRPML1[110], that facilitate the coupling of lysosomes to dynein. Furthermore, findings from studies conducted in Drosophila have demonstrated that Arl8 has the ability to interact with the ortholog of RILP[48]. An important question arising from these discoveries is how and when this collection of proteins is engaged to drive

lysosomal transport. It is plausible that these adapters interact differently with distinct subpopulations of lysosomes. Another possibility is that they play tissue-specific roles. Moreover, in the mammalian system, where approximately forty-five kinesins coexist with one cytoplasmic dynein, it may be necessary to regulate the balance between these molecular motors to reinforce dynein-mediated lysosomal positioning. Furthermore, recent research has unveiled a sequential recruitment of proteins that recruit dynein during retrograde transport of autophagosomes in axons[111]. Given these various possibilities, further studies are warranted to explore whether there is competitive recruitment of these adapters, which could influence downstream signaling events. Additionally, understanding the physiological cues that dictate the recruitment of these adapters to lysosomes is also necessary to gain a comprehensive understanding of the process.

## Methods

### Cell lines
HeLa and HEK-293T cells were from ATCC (CCL-2 and CRL-1573).

### Cell culture
Cell lines were cultured in DMEM high-glucose (GE Healthcare cat# SH30081.01) containing 10% bovine calf serum (GE Healthcare cat# SH30072.03), 2 mM L-glutamate (Wisent cat# 609065), 100 IU penicillin and 100 μg/ml streptomycin (Wisent cat# 450201). Serum starvation media: DMEM high-glucose containing 2 mM L-glutamate, 100 IU penicillin and 100 μg/ml streptomycin. Cell lines were tested for mycoplasma contamination routinely using the mycoplasma detection kit (Lonza; cat# LT07-318). Earle's Balanced Salts (EBSS) was purchased from Sigma (E2888).

### DNA constructs
All 60 GFP-Rab constructs were gifts from M. Fukuda (Tohoku University)[112–115]. The following constructs were custom synthesized by SynBio technologies: mito-mScarlet-DENND6A (DENND6A is human; vector- pmScarlet-i_C1, addgene 85044), mito-mScarlet (vector-pmScarlet-i_C1, Addgene 85044), DENND6A-GFP (vector- pEGFP-N1), GFP (vector- pEGFP-N1), PEX3(amino acids 1-42)-FKBP-mCherry (FK506-binding protein domain (FKBP) fragment was synthesized as per addgene 46944; vector- pEGFP-C1), FRB-GFP (FRB was amplified from addgene 59352; vector- pEGFP-N1), DENND6A-FRB-EGFP (vector-pEGFP-N1), GST-Rab34 (vector pGEX-6p-1), GST-Rab34-Q111L (vector-pGEX-6p-1), GST-Rab34-T66N (vector- pGEX-6p-1), T7-RILP (vector- pET-24a(+)), mCherry-Rab34 (vector- pEGFP-C1; replaced EGFP with custom synthesized Arl8b-mCherry), Arl8b-mCherry (vector- pEGFP-C1; replaced EGFP with mCherry), GST-Arl8b QL (vector-pGEX-6p-1), GST-Arl8b TN (vector- pGEX-6p-1), mito-mScarlet-Arl8b QL (vector- pmScarlet-i_C1), mito-mScarlet-Arl8b QL (vector- pmScarlet-i_C1), Arl8b QL-FLAG (vector- pEGFP-C1, EGFP was replaced by custom synthesized Arl8b QL-FLAG), Arl8b TN-FLAG (vector- pEGFP-C1, EGFP was replaced by custom synthesized Arl8b QL-FLAG), GFP-Rab34 Q111L (pEGFP-C1), mScarlet-Rab34 (vector- pmScarlet-i_C1), GFP-DENND6A (vector- pEGFP-C1). GFP-DENND2B is described previously[72].

### Generation of DENND6A KO HeLa line
Three guide RNAs (gRNAs) (Target 1-42 AAGGCCGTTGGACGAAGCGG; Target 2-47 CGTTGGACGAAGCGGTGGCA; Target 3-675 TTAC-CACCCCCATGATTGGC) were obtained from Applied Biological Materials (catalogue number: 19727111) and separately cloned into pLenti-U6-sgRNA-SFFV-Cas9-2A-Puro to increase the chances of generating a KO line. Lentivirus-based delivery of gRNA and Cas9 was used to KO DENND6A from HeLa cells. One 15-cm plate containing 107 HEK-293T cells were transfected with 7.5 μg of each gRNA constructs, 15 μg of psPAX2 (obtained from S. Pfeffer), and 7.5 μg of pMD2 VSV-G (obtained from S. Pfeffer) using calcium phosphate. At 8 hours post-transfection, the culture medium was replaced with collection medium [15 ml per

plate; regular medium supplemented with 1× nonessential amino acids (Gibco) and 1 mM sodium pyruvate (Gibco)]. The medium was collected at 24 and 36 hours and replaced with fresh medium (15 ml per plate) with each collection. The collected medium at 24 hours was stored at 4 °C until the last collection. The collected culture media were then filtered through a 0.45-μm α-polyethersulfone (PES) membrane. A total of $5 \times 10^4$ HeLa cells were seeded in one well of a 24-well plate. The next day, regular culture media were replaced with the filtered supernatant containing lentivirus (2 ml each well) and incubated for 48 hours. Following incubation, puromycin-resistant cells were selected with puromycin (2.5 μg/ml) in the culture medium for 48 hours. After selection, cells were isolated by clonal dilution. Following the expansion of selected colonies, KOs were confirmed by western blot.

### Transfection
HeLa cells were transfected using the jetPRIME Transfection Reagent (Polyplus) according to the manufacturer's protocol. HEK-293T cells were transfected using calcium phosphate.

### Small interfering RNA–mediated knockdown
HeLa cells were plated at ~80% confluency. Cells were transfected using jetPRIME Transfection Reagent (Polyplus) according to the manufacturer's guidelines and were used 48 hours after knockdown. Control small interfering RNA (siRNA) (ON-TARGETplus; D-001810-10-20), Rab34 siRNA-targeting genome pool (SMARTpool:ON-TARGETplus; L-009735-00-0010), Arl8a siRNA-targeting genome pool (SMARTpool:ON-TARGETplus; L-016577-01-0010), Arl8b siRNA-targeting genome pool (SMARTpool:ON-TARGETplus; L-020294-01-0010), Dynein (DIC) siRNA-targeting genome pool (SMARTpool:ON-TARGETplus; L-006828-00-0010), and RILP siRNA-targeting genome pool (SMARTpool:ON-TARGETplus; LQ-008787-01-0010) were purchased from Dharmacon/Horizon Discovery.

### Antibodies and reagents
Mouse monoclonal Flag (M2) (F3165) antibody is obtained from Sigma-Aldrich [Western blot (WB)- 1:5000]. Rabbit polyclonal GFP (A-6455) is obtained from Invitrogen (WB-1:5,000), and rat monoclonal HSC70 antibody (WB-1:10,000) is obtained from Enzo (ADI-SPA-815-F). Alexa Fluor 488 (rabbit- A11088; mouse- A11001)–, Alexa Fluor 568 (rabbit-A10042; mouse- A11031)–, and Alexa Fluor 647 (rabbit- A21245; mouse-A31571)–conjugated rabbit or mouse secondary antibodies are from Invitrogen. Anti-LAMP1 mouse antibody (H4A3) is from Developmental Studies Hybridoma Bank [Immunofluorescence (IF)- 1:500, WB- 1:1000], anti-LAMP1 rabbit antibody (IF-1:200) is from Cell Signaling Technology (D2D11), anti-DENND6A antibody (PA5-66508) is from Thermo Fisher Scientific (WB-1:1000), anti-GAPDH antibody (TA802519) is from Origene (WB-1:5000), anti-HA antibody (C29F4; 3724 S) is from (WB-1:1000) Cell Signaling, anti-VDAC antibody (4866 S) is from Cell Signaling (WB-1:1000), anti-Catalase antibody (D4P7B; 12980 S) is from (WB-1:1000) Cell Signaling, anti-Rab34 antibody (sc-376710) is from Santa Cruz Biotechnology (WB-1:1000), anti- Cytoplasmic Dynein Intermediate Chain antibody (904901) is from BioLegend (WB-1:1000), anti-Arl8a antibody (17060-1-AP) is from Proteintech (WB-1:1000), anti-Arl8b antibody (13049-1-AP) is from Proteintech (WB-1:1000), anti-LC3B antibody (3868) is from Cell Signaling (WB-1:1000; IF-1:500), anti-RILP antibody (13574-1-AP) is from Proteintech (WB-1:1000), anti-p62 antibody (MAB80281) is from R&D Systems (WB-1:1000), anti-p150 (Glued) antibody (610474) is from BD Transduction (WB-1:1000), anti-ULK1 (8054) antibody is from Cell Signaling (WB-1:1000), anti-Phospho-ULK1 (6888) antibody is from Cell Signaling (WB-1:1000), anti-GM130 antibody (610822) is from BD Transduction (IF-1:500).

### Imaging
HeLa cells were plated on poly-l-lysine–coated coverslips. Cells were fixed with warm 4% paraformaldehyde for 10 min at 37 °C,

permeabilized for 5 min in 0.1% Triton X-100 in phosphate-buffered saline (PBS), and blocked for 1 hour in 1% bovine serum albumin in PBS (blocking buffer). Coverslips were incubated in a blocking buffer containing diluted primary antibodies and incubated overnight at 4 °C. Cells were washed three times for 10 min with blocking buffer and incubated with corresponding Alexa Fluorophore–conjugated secondary antibodies diluted 1:1000 in blocking buffer for 1 hour at room temperature. Cells were washed three times for 10 min with blocking buffer and once with PBS. Coverslips were mounted on a microscopic slide using fluorescence mounting medium (Dako, catalog no. S3023).

**Live cell imaging.** A total of 50,000 HeLa cells were initially seeded on PLL-coated 35-mm No. 1.5 glass-bottom dishes (MatTek Corporation). These cells were subsequently transfected with either GFP alone or DENND6A-GFP. 24 hours for post-transfection, cells were labeled with 100 nM Lysotracker Red (DND-99; Invitrogen) for 1 h at 37 °C and live cell imaging was conducted using a Zeiss LSM880 with AiryScan (single frame per second). This imaging setup included a humidified environmental chamber that was maintained at a temperature of 37 °C with 5% $CO_2$.

Confocal imaging was performed using a Leica SP8 laser scanning confocal microscope. Images of lysosomes marked by LAMP1 are presented as maximum intensity projections. Image analysis was done using ImageJ. All the images were prepared for publication using Adobe Photoshop (adjusted contrast and applied 1-pixel Gaussian blur) and then assembled with Adobe Illustrator.

### 3D Structure Illumination Microscopy (3D-SIM)

3D-SIM images were acquired using LSM880-Elyra PS1 super-resolution microscopy (Zeiss, Germany) with the Plan Apochromat 100x (NA 1.46) oil-immersion objective. 488, 561 or 640 nm lasers were used to excite corresponding fluorophore. 3D stacks were recorded with an interval of 120 – 150 nm. A total of 15 images with 3 different phases X 5 different angles were captured for each XY plane. Reconstruction of the 3D SIM images was performed using ZEN Elyra software (gray version).

### Degree of colocalization

The degree of colocalization was quantified using Imaris Software at the Analysis Workstation of Advanced BioImaging Facility (McGill). Channel corresponding to lysosome was masked such that only the lysosomal area is evaluated. Furthermore, images were thresholded automatically using the Imaris algorithm, and the Pearson correlation coefficient was calculated between the two indicated fluorescent signals.

### Screening of 60 GFP-Rabs

HeLa cells (9000 cells per 100 ml of culture medium) were seeded in each well of a 96-well plate CellCarrier-96 Ultra Microplates (6055302, PerkinElmer). Cells were cotransfected with 100 ng each of individual GFP-Rabs and mito-mScarlet-DENND6A or mito-mScarlet using jet-PRIME. At 24 hours post-transfection, cells were fixed with 4% paraformaldehyde, washed three times with PBS.

Each transfected well of the 96-well plate was divided into grids and between 70 and 100 images were acquired by the Opera Phenix HCS microscope using a 63x objective. A qualitative assessment of localization of GFP-Rabs at the mitochondria were performed on all the images by eye estimation. Furthermore, all potential hits were further confirmed by imaging each pair using Leica SP8, which provides higher resolution.

### Protein purification

GST-Rab34 QL, GST-Rab34 TN, T7-RILP proteins were expressed in Escherichia coli BL21 (500 μM isopropyl β-d-1-thiogalactopyranoside; Wisent Bioproducts; at room temperature for 16 hours) and purified

using standard procedure in tris buffer [20 mM tris (pH 7.4), 100 mM NaCl, 10 mM MgCl2, and 1 mM dithiothreitol] supplemented with protease inhibitors.

### Biochemical assays

**Preparation of cell lysate for immunoblot.** To analyze protein levels, cells were lyzed in phospho-lysis buffer [20 mM Hepes, 100 mM NaCl, 1 mM dithiothreitol, 1% Triton X-100 (pH 7.4)], supplemented with protease inhibitors [0.83 mM benzamidine, 0.20 mM phenylmethylsulfonyl fluoride, aprotinin (0.5 mg/ml), and leupeptin (0.5 mg/ml)] and Phosphatase inhibitor cocktail (Cell Signaling, 5870). Cell lysates were centrifuged at 21,130 g for 10 min at 4 °C, and the supernatant was resolved by SDS–polyacrylamide gel electrophoresis (PAGE) and processed for Western blotting.

**Lyso-IP.** The isolation of lysosomes and mitochondria was performed following the LysoIP protocol[71]. Tmem192-3xHA (Addgene #102930), Tmem192-2xFlag (Addgene #102929) were gifts from Dr. David Sabatini. HEK-293 cells were transfected with either Tmem192-3xHA (HA-Lyso cells) or with TMEM192-2xFlag (Control-Lyso cells).

Approximately 35 million cells were used for each immunoprecipitation. Cells were quickly rinsed twice with cold PBS and scraped in 2 ml of KPBS (136 mM KCl, 10 mM KH2PO4, pH 7.25 and protease inhibitor) and centrifuged at 1000 x g for 2 min at 4 °C. Pelleted cells were resuspended in 900 μl of KPBS and gently homogenized with 20 strokes in a 2 ml hand-held homogenizer. The homogenate was centrifuged at 1000 x g for 2 min at 4 °C. The supernatant was collected (equivalent to 2.5%) and was reserved and run on immunoblot as the starting material. The remaining supernatant was incubated with 150 μl of KPBS prewashed anti-HA magnetic beads (Thermo Fisher Scientific cat# 88837) on a gentle rotator shaker for 3 min. Beads were gently washed 3 times with 1 ml of KPBS using a DynaMag-2 magnet (Thermo Fisher Scientific cat# 12321D). Beads were resuspended in 1 ml of KPBS and transferred to a new tube. KPBS was removed and beads were incubated in resuspension buffer (20 mM HEPES, 150 mM NaCl, 1% Triton X-100, pH 7,4 + protease inhibitor). Supernatant was collected and run on an immunoblot.

**Effector pull-down assay.** Cells were gently washed with PBS, lyzed in lysis buffer [20 mM Hepes, 100 mM NaCl, 20 mM MgCl2, 1 mM dithiothreitol, and 1% Triton X-100 (pH 7.4)] supplemented with protease inhibitors, and incubated for 20 min on a rocker at 4 °C, and the lysates were centrifuged at 305,000 g for 15 min at 4 °C. For GST or T7 pull-down experiments, supernatants were incubated with GST fusion proteins precoupled to glutathione-Sepharose beads or T7 tagged protein coupled to T7•Tag Monoclonal Antibody (covalently cross-linked agarose beads) for 1 hour at 4 °C. GST or T7 beads attached to the fusion proteins were washed three times with the same lysis buffer, eluted in SDS-PAGE sample buffer, resolved by SDS-PAGE, and processed for immunoblotting.

### GEF assay

GST-tagged Rab34 was expressed in BL21 bacteria and purified using standard procedure. GST tags were cleaved with PreScission protease by overnight incubation at 4 °C and then the cleaved GTPases were exchanged into GEF loading buffer (20 mM HEPES, pH 7.5, and 100 mM NaCl). Flag-tagged DENND6A was expressed in HEK-293T cells. At 24 h post-transfection, cells were collected in lysis buffer (20 mM HEPES, 100 mM NaCl, 1% Triton X-100 and supplemented with protease inhibitors) and incubated for 30 min at 4 °C. Lysates were centrifuged for 15 min at 21,000 g and the supernatants were incubated with monoclonal FLAG (M2) antibody and protein G-Sepharose beads for 2 h at 4 °C. Beads were washed in GEF incubation buffer (20 mM HEPES, pH 7.5, 100 mM NaCl, and 5 mM MgCl2) and the immunoprecipitated Flag-DENND6A was immediately used for in vitro GDP/GTP exchange assay.

For the GDP/GTP exchange assays, 4 µM of purified Rab34 was loaded with 20 µM GDP by incubation for 10 min at 30 °C in GEF loading buffer containing 5 mM EDTA. Loaded GDP was stabilized using 10 mM MgCl2, and then samples were incubated for 10 min at 30 °C. Exchange reactions were performed at room temperature in 90 µl of total volume containing 0.4 µM preloaded Rab34, immuno-precipitated Flag-DENND6A, 0.5 mg/ml bovine serum albumin, 5 µM GTPγS, 0.2 mCi/mmol [$^{35}$S]GTPγS (PerkinElmer Life Sciences), 0.5 mM dithiothreitol, and 5 mM MgCl2 in GEF incubation buffer. At the indicated time points, 15 µl of the reaction was removed, added to 1 ml of ice-cold wash buffer (20 mM HEPES, pH 7.5, 100 mM NaCl, 20 mM MgCl2), and passed through nitrocellulose filters. The filters were washed with 5 ml of cold wash buffer and then counted using a liquid scintillation counter (Beckman Coulter LS6500 scintillation counter).

### Immunoblot
Lysates were run on large polyacrylamide gels and transferred to nitrocellulose membranes. Proteins on the blots were visualized by Ponceau staining. Blots were then blocked with 5% milk in tris-buffered saline with 0.1% Tween 20 (TBST) for 1 hour followed by incubation with antibodies O/N at 4 °C diluted in 5% milk in TBST. The next day, blots were washed three times with TBST. Then, the peroxidase-conjugated secondary antibody was incubated in a 1:5000 dilution in TBST with 5% milk for 1 hour at room temperature followed by washes.

### Autophagic flux experiment
Cells were washed with PBS for 5 times. Subsequently, the cells were treated as specified in the experiment, with or without the V-ATPase inhibitor Bafilomycin A1 (100 nM, obtained from Sigma-Aldrich), in the presence of EBSS for 3 h. Following treatment, cells lysates were analyzed using immunoblot.

**LC3 staining analysis**. Each image frame was opened individually in ImageJ. The cell perimeter was defined by thresholding equivalent saturated images. Then a region of interest (ROI) was drawn using the custom region draw function around the entire cell. These regions were then measured for mean intensity. The resulting data were plotted using GraphPad Prism software.

### Endocytosis (DQ-BSA) assay
To assess endocytic cargo degradation, 40,000-50,000 WT or DENND6A KO cells were seeded on PLL-coated coverslips within a 4-chambered dish. Subsequently, the cells were incubated in culture media containing 20 µg/mL of DQ-BSA (D12051, Invitrogen) for 2 hours at 37 °C. After the incubation, the culture media was removed, and the cells were fixed with 4% paraformaldehyde at room temperature, followed by DAPI staining. Finally, coverslips were mounted onto microscopic slides using fluorescence mounting medium (Dako, catalog no. S3023).

### Lysosomal distribution
To analyze lysosomal distribution, cells were either untreated or starved for 4 h in the presence of EBSS. Following treatment, cells were fixed and processed for imaging using 63x objective at Leica SP8.

**Quantification using Fiji software**. The cell perimeter was defined by thresholding equivalent saturated images. The image analysis was done in Fiji. Each cell was manually traced. The cell outlines were consecutively reduced in size by a fixed length of 10 times. The integrated density is measured respectively for each area. A percentage of the integrated density of the area over the total cell area is calculated for lysosomal distribution within each cell. Upon plotting the data points, the GraphPad Prism software was employed to fit a centered 6th order polynomial using nonlinear regression. In order to analyze the

statistical significance in the distribution profiles, the Extra Sum of F-squares test was utilized.

### Statistics
Graphs were prepared using GraphPad Prism software. For all data, the normality test was performed before determining the appropriate statistical test. For normally distributed data, comparisons were made using either two-tailed unpaired t test or ANOVA. For non-normally distributed data, comparisons were made using non-parametric tests. All data are shown as the mean +/- SEM with $P < 0.05$ considered statistically significant, and the reported experimental data was obtained through either three experimental repeats or by considering the indicated number of cells mentioned in the figure legends. All images (microscopy images or western blots) are representative of 3 replicates.

### Reporting summary
Further information on research design is available in the Nature Portfolio Reporting Summary linked to this article.

## Data availability
The manuscript and Supplementary Information file contain all the pertinent data that underpins the findings of this study. Raw data including uncropped western blots are accessible in the accompanying Source Data file, which is included with this paper. Source data are provided with this paper.

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

## Acknowledgements

We thank Jacynthe Philie, Maryam Fotouhi and Philip Thorne for their excellent technical assistance. We acknowledge the Neuro Microscopy Imaging Centre and Advanced BioImaging Facility at McGill University. Graphical representation of the findings was created by BioRender. This work was supported by a Canadian Institutes of Health Research Foundation Grant to PSM. RK is supported by a studentship from ALS Canada and the Canada First Research Excellence Fund, awarded to McGill University for Healthy Brains for Healthy Lives. VF was supported by a fellowship from the Fonds de recherche du Quebec – Sante (FRQS). GK was supported by FRQS and a Jeanne Timmins Costello Fellowship. PSM is a Distinguished James McGill Professor and a Fellow of the Royal Society of Canada.

## Author contributions

R.K. conceived the experiments. R.K., V.F., M.K., A.A., G.K. and E.B. performed the experiments. R.K. and P.S.M analyzed the experiments. R.K. and P.S.M. wrote the manuscript. P.S.M supervised and funded the work.

## Competing interests

The authors declare no competing interests.

## Additional information

**Peer review information** : *Nature Communications* thanks Hiroshi Hashimoto and the other, anonymous, reviewer(s) for their contribution to the peer review of this work. A peer review file is available.

