## [Peer Review File · Nature Communications]

REVIEWER COMMENTS

Reviewer #1 (Remarks to the Author):

Kamar et al. explore the role of the poorly characterized DENND6A protein in regulating lysosomal positioning and autophagy. Analysis of proteomic data suggests the possibility that DENND6A associates with lysosomes. This is tested in overexpression experiments, which show a punctate and reticulate DENND6A-GFP pattern that partially overlaps with the lysosomal membrane protein LAMP1. Overexpression of DENND6A-GFP promotes the juxtannuclear clustering of lysosomes, and conversely, the CRISPRKO of DENND6A causes their dispersion. In accordance with this, targeting DENND6A to peroxisomes also promotes their juxtannuclear clustering. Since other DENN domain proteins have been shown to act as GEFs for Rab GTPases, a panel of Rab GTPases is screened in a mitochondrial targeting assay. This appears to indicate a very broad potential substrate specificity; however, the authors focus on Rab34 because it has been previously implicated in the control of lysosome positioning. There is no known GEF for Rab34, so DENND6A could be a candidate. To explore this, the authors present a series of biochemical and cell-based assays that make a strong case that DENND6A activates Rab34 and enables it to bind its dynein binding effector RILP. Furthermore, DENND6A-induced lysosome clustering is Rab34, RILP, and dynein dependent. The authors go on to provide evidence that DENND6A is an effector for Arl8b – a small GTPase also implicated in the transport of lysosomes. Functionally, DENND6A loss impairs the starvation-induced clustering of lysosomes and autophagic flux.

The authors present a model where Arl8 recruits DENND6A to lysosomes, where it recruits and activates Rab34. This, in turn, recruits RILP and dynein to drive transport to the perinuclear region.

Overall, I felt that the manuscript is well-written and easy to read, containing some important and novel findings. Firstly, it convincingly implicates DENND6A in lysosome positioning and, secondly, provides strong data suggesting that it acts as a heretofore elusive GEF for Rab34, controlling lysosomal positioning via that pathway. However, I do think there is a major issue that would need to be addressed before the manuscript is suitable for publication.

It appears that there is substantial divergence between the data and the model presented. The data in Figure 1 do not support the proposition that DENND6A localizes to lysosomes. A correlation coefficient of <0.1 would normally indicate that two proteins do not colocalize. While this is 'significantly' increased compared to GFP, one might expect this from any punctate or reticulate staining pattern in the juxtannuclear region of the cell. At best, the authors can state that DENND6A-GFP partially overlaps with LAMP1 staining. It is unclear whether the authors wish to claim that DENND6A localizes to lysosomes (see results section heading) or that distinct 'puncta and tubular structures of DENND6A were associated with LAMP1 vesicles' and 'wrapped around the LAMP1 vesicles.' To me, there is an important distinction

between these two possibilities. The former is consistent with the model presented in Figure 9. The latter is not.

Building on this point, prior studies have indicated that Rab34 is a Golgi-localized protein (Wang and Hong (2002) 10.1091/mbc.e02-05-0280, Goldenberg et al. (2007) 10.1091/mbc.E06-11-0991, and Starling et al. EMBO Reports (2016) 10.15252/embr.201541382) and acts on lysosomes from the Golgi compartment (Wang and Hong (2002) 10.1091/mbc.e02-05-0280). The data presented in Figure 5F in the present study appear consistent with the above studies, and (in this reviewer's interpretation) the non-lysosomal localization of DENND6A presented here may also be consistent with this.

It is important that the authors adapt their manuscript and model to include consideration of this possibility or provide stronger evidence to refute it. This might, for example, include endogenous immunostaining of DENND6A clearly 'on lysosomes,' rather than associated with another compartment, with evidence that endogenous Rab34 is recruited to lysosomes in a DENND6A-dependent manner (I am not aware if Rab34 comes up in any lysosome proteomic studies) and that they localize together with RILP.

Reviewer #2 (Remarks to the Author):

In the manuscript by Kumar et al., the authors examine the role that DENND6A plays in lysosomal positioning via interaction with ARL8 and Rab34 activation, which the authors propose leads to retrograde trafficking of lysosomes from the periphery to the perinuclear region. This is an interesting paper that adds critical details to the mechanisms of lysosomal trafficking and assigns a role for DENND6A in this process. Additionally, the authors show that disruption of DENND6A results in reduced autophagic flux upon amino acid starvation, adding to pathway relevance. Regardless, there are a few points on DENND6A localization that require clarification, as well as some additional controls – as detailed below.

Main Points:

- 1) Where is the majority of DENND6A localized? While the authors convincingly show that a small pool of DENND6A is at lysosomes, most of it is not. This makes it less clear as to what is happening to these other compartments upon manipulation of DENND6A levels. Is it possible that disruption of these other compartments could be indirectly affecting lysosomal function and/or autophagy? Is DENND6A on earlier endosomal compartments?

2) Related to the above, the authors use exogenous GFP-DENND6A for localisation, which could be mislocalized – is it possible to visualize endogenous DENND6A? What is the expression of GFP-DENND6A relative to endogenous DENND6A?

3) In Figure 7F, loss of ARL8a/b results in a peripheral relocalization of DENND6A, but not of LAMP1 (as detailed in insets) – I was slightly confused here as to why lysosomes were not more peripheral too (as they cannot now recruit DENND6A), so maybe more explanation is needed? Likewise, as mentioned in point 1 – what is the identity of the DENND6A peripheral compartment here?

4) The autophagy data in Figure 8 does indeed suggest reduced levels of autophagy in the absence of DENND6A, but it was not clear why total levels of LC3B were lower in these cells – this does not fit with the authors argument that the defect is at the level of autophagosome-lysosome fusion (if fusion was blocked then there should be more LC3-II, as with bafilomycin). The authors should blot for another marker such as p62/SQSTM1, which also accumulates when autophagy is blocked.

5) Is the autophagy block related to the defect in lysosomal positioning? It could be beneficial to expand the IF data shown in Figure 8E to include the EBSS conditions +/- Baf: are autophagosomes also more peripheral in absence of DENND6A or is it just the lysosomes?

6) Related, maybe autophagy initiation is also blocked in absence of DENND6A? They could look at WIPI2 puncta formation or ULK1 substrate phosphorylation. This may relate to defect in mTOR signalling, especially given the changes in lysosome localization.

7) Finally, the authors could do with some extra controls. Given the phenotypes in the KO clones, can these be rescued by re-expressing WT DENND6A? Also, what happens if another DENND protein is overexpressed – does this result in a similar perinuclear clustering phenotype, or is it exclusive to DENND6A?

Reviewer #3 (Remarks to the Author):

In the present work, Kumar et al. have investigated the role of DENN domain protein family member DENND6A in retrograde transport and positioning of LAMP1-positive compartments, collectively referred to as lysosomes. The authors show that DENND6A partially localizes to lysosomes, and its overexpression results in juxtannuclear positioning of lysosomes. Conversely, depletion of DENND6A leads to the peripheral distribution of lysosomes. Further, using the FRB-FKBP-rapamycin system, the authors show that DENND6A is sufficient to cause clustering of target organelles. DENND6A is a member of the DENN domain family of proteins, which constitute the largest family of Rab GEFs. The authors have used a relocalization approach (targeting of DENND6A to mitochondria) to identify potential Rab binding partners for DENND6A. The screening resulted in the identification of 20 Rabs, from which the authors have further characterized the interaction of DENND6A with Rab34, a golgi- and endolysosomal-

localized Rab GTPase. The author shows that DENND6a exhibits GEF activity towards Rab34, and Rab34 is required for DENND6A-mediated lysosome positioning. Previously, Rab34 was shown to mediate juxtannuclear clustering of lysosomes and required interaction with its effector, RILP, a dynein adaptor. Using a recombinant protein pulldown approach, the authors show that DENND6A interacts with the Rab34-RILP-DIC (the intermediate chain of the dynein motor machinery) complex.

Previously, using a BioID-based interactome, DENND6A was reported as a potential binding partner of the lysosome-localized small GTPase Arl8b. In the present manuscript, the authors confirmed this interaction and showed that DENND6A acts as an effector of Arl8b and colocalizes with Arl8b in LAMP1-positive compartments. Arl8b regulates the positioning of lysosomes in cells by recruiting specific effectors for the loading of dynein or kinesin motor proteins on lysosomes. Interestingly, the authors report that silencing of Arl8a/b alters the perinuclear distribution of DENND6A-compartments. Finally, the authors have shown that DENND6A regulates the nutrient-dependent positioning of lysosomes and positively regulates autophagic flux.

Overall, this is an interesting study, as the authors have characterized the role of DENND6A in lysosome positioning, which was previously unknown. However, the study in its present form falls short of the impact. Particularly, some of the major conclusions drawn by the authors need more experimental evidence (with the addition of key controls and quantification), which is currently missing in the manuscript. Below are specific concerns about some of the experiments and their presentation.

Major concerns:

1) In Fig. 1B-E, colocalization of overexpressed DENND6A-GFP with LAMP1-positive compartments is quite low (PCC value of ~ 0.15). Representative images also show DENND6A-GFP localization to tubular and non-LAMP1-positive structures. First, the authors should reassess the colocalization by including more cell numbers from at least three independent experiments, as the data plotted is only from 25 cells, as per the figure legend. The identity of non-LAMP1-positive DENND6A compartments should be established by the authors. Also, authors should confirm whether the C-terminal GFP-tagging of DENND6a is not causing mislocalization, and therefore, N-terminal GFP-tagging or the use of small tags such as FLAG and HA should be explored. If possible, the authors should try to visualize the endogenous distribution of DENND6A in HeLa cells.

2) The authors should confirm that DENND6A-GFP is a functional protein and can rescue the effects of loss of DENND6A on lysosome positioning as shown in Figs. 2E-F. Also, the effect of lysosome positioning should be rescued with a GEF activity-deficient mutant of DENND6A if the authors need to establish that DENND6A GEF activity is required for lysosome positioning.

3) Does DENND6A-GFP localize to acidic and degradative lysosomes as well? This is important to determine as both the non-degradative and degradative populations will be marked by LAMP1. Preferably, authors should do these experiments in live cells using dyes such as LysoTracker and/or SiR-lysosome or Magic Red. These dyes generally show diffuse localization in fixed cells.

4) In the context of the above point, the authors should report the effect of DENND6A overexpression on Rab7-positive compartments. Does DENND6A specifically reposition Arl8b-positive endolysosomes and not Rab7-marked endolysosomes? This will be important as RILP is a well-known effector of Rab7.

5) As shown in Fig. 3, DENND6A targeted to peroxisomes is sufficient to cause their perinuclear clustering. As in the author's model (Fig. 9), DENND6A does not directly bind the dynein motor but via the RAB34-RILP complex. The authors should test whether DENND6A-mediated peroxisomal clustering is Rab34- and RILP-dependent.

6) In Fig. 5C, authors should include a control panel showing levels of another endolysosomal Rab protein (such as Rab7 since it was not recruited by DENND6A as per the screening data mentioned in Fig. 4) to strengthen the conclusion that overexpression of DENND6A increases the active levels of Rab34 in the cell. Similarly, the inclusion of a GEF activity-defective mutant of DENND6A in Fig. 5E should be tested.

7) Based on the results shown in Figs. 5J–K, the authors have concluded that Rab34 is unable to mediate perinuclear positioning of lysosomes in the absence of DENND6A. However, from the images shown, it seems Rab34 is still able to mediate the perinuclear positioning of LAMP1-positive compartments in DENND6A KO cells (compare the positioning of LAMP1-positive compartments in untransfected vs. Rab34-transfected cells in DENND6A KO HeLa cells). Can the authors clarify this point? To establish that Rab34 is downstream of DENND6A-mediated GEF activity, authors can express a constitutively active version of Rab34 and check whether that rescues the effect of DENND6A KO on lysosome positioning.

8) In Figs. 5F and 5J, Rab34 localization looks very perinuclear and resembles Golgi staining. Have the authors checked Rab34 and DENND6A-GFP colocalization with Golgi? The image in Fig. 5F should be again performed with SIM imaging, as it is not clear whether mCherry-Rab34 and DENND6A-GFP are colocalized with LAMP1. Also, authors should analyze the localization of Rab34 (endogenous if possible) on lysosomes in DENND6A KO cells.

9) The authors report an exciting result wherein DENND6A is shown to act as an effector of Arl8b. However, this was established based on a GST-pulldown assay (Fig. 7A). The authors should strengthen this data by performing additional assays. Does Arl8b interact with DENND6A in cells in a GTP-dependent

manner? This can be tested by performing co-immunoprecipitation experiments using GTP/GDP-locked mutants of Arl8b. If DENND6A is an Arl8b effector, then mito-Arl8b (mitochondrial localized Arl8b) should recruit it on the membrane. Further, DENND6A localization to lysosomes should be reduced in Arl8b-silenced cells. This could be tested by performing confocal imaging and biochemical analysis using the Lyso-IP approach. In the same experiments, levels of Rab34 on lysosomes can be determined.

10) The authors should test if DENND6A competes with SKIP for binding to Arl8b in order to reposition lysosomes towards the cell center. Recently, RUFY3 was shown to act as an effector of Arl8b to mediate the perinuclear positioning of lysosomes. How are these effectors of Arl8b (RUFY3 and DENND6A) mediating juxtannuclear repositioning of lysosomes? Is RUFY3 also present in complex with DENND6A? Is RUFY3 still able to relocalize Arl8b-positive lysosomes in DENND6A KO cells?

11) As per the authors' model (Fig. 9), Arl8b recruits DENND6A, which in turn recruits Rab34 on lysosomes. This is not shown in the study. Does DENND6A directly interact with Arl8b? Identification of the Arl8b-binding site on DENND6A and further rescue with an Arl8b-binding defective mutant of DENND6A will further strengthen the claim that Arl8b recruits DENND6A to mediate Rab34 GDP-GTP exchange on lysosomes.

12) It is surprising that the authors have not explored the role of DENND6A-mediated lysosome positioning on lysosome fusion with late endosomes and thereby cargo degradation. These experiments will signify the impact of DENND6A-mediated positioning of lysosomes on their function in cargo degradation.

Minor points:

1) If possible, the authors should test the effect of Arl8b overexpression or knockdown on Rab34 and DENND6A interactions.

2) In Fig. 6A, using a recombinant RILP protein in a pulldown assay, the authors found the presence of a complex containing DENND6A, Rab34, dynein, and RILP, suggesting that they function together in the same pathway. The authors should test the presence of this complex in vivo by co-immunoprecipitation and, if possible, probe for the presence of dynactin subunits in the complex.

3) The results shown in Fig. 7F on LAMP1 positioning in Arl8a/b siRNA are somewhat confusing. Several studies have reported that Arl8a/b silencing leads to perinuclear positioning of lysosomes in HeLa cells. However, the LAMP1 staining shown in control vs. Arl8a/b siRNA seems to be the same. Can the authors clarify this point?

5) In the experiments shown in Fig. 8C, the authors should determine the levels of LC3B-I and p62 in order to better evaluate the effect of DENND6A silencing on autophagic flux.

6) In the material and methods section, information on the antibody used in immunofluorescence for LC3 should be mentioned. Also, in the figure legends, what the yellow arrow depicts should be mentioned.

We thank the reviewers for their valuable and positive feedback. We have addressed all comments of all reviewers, in many cases, through the addition of new data. In the submitted manuscript file, major modifications have been highlighted in **blue**. Please find below a detailed response (in **blue**) to the reviewer's comments:

Reviewer #1:

Kamar et al. explore the role of the poorly characterized DENND6A protein in regulating lysosomal positioning and autophagy. Analysis of proteomic data suggests the possibility that DENND6A associates with lysosomes. This is tested in overexpression experiments, which show a punctate and reticulate DENND6A-GFP pattern that partially overlaps with the lysosomal membrane protein LAMP1. Overexpression of DENND6A-GFP promotes the juxtannuclear clustering of lysosomes, and conversely, the CRISPRKO of DENND6A causes their dispersion. In accordance with this, targeting DENND6A to peroxisomes also promotes their juxtannuclear clustering. Since other DENN domain proteins have been shown to act as GEFs for Rab GTPases, a panel of Rab GTPases is screened in a mitochondrial targeting assay. This appears to indicate a very broad potential substrate specificity; however, the authors focus on Rab34 because it has been previously implicated in the control of lysosome positioning. There is no known GEF for Rab34, so DENND6A could be a candidate. To explore this, the authors present a series of biochemical and cell-based assays that make a strong case that DENND6A activates Rab34 and enables it to bind its dynein binding effector RILP. Furthermore, DENND6A-induced lysosome clustering is Rab34, RILP, and dynein dependent. The authors go on to provide evidence that DENND6A is an effector for Arl8b – a small GTPase also implicated in the transport of lysosomes. Functionally, DENND6A loss impairs the starvation-induced clustering of lysosomes and autophagic flux.

The authors present a model where Arl8 recruits DENND6A to lysosomes, where it recruits and activates Rab34. This, in turn, recruits RILP and dynein to drive transport to the perinuclear region.

Overall, I felt that the manuscript is well-written and easy to read, containing some important and novel findings. Firstly, it convincingly implicates DENND6A in lysosome positioning and, secondly, provides strong data suggesting that it acts as a hereto elusive GEF for Rab34, controlling lysosomal positioning via that pathway. However, I do think there is a major issue that would need to be addressed before the manuscript is suitable for publication.

Response to reviewer 1: We thank the reviewer for their positive evaluation of our study. We have addressed the reviewer's comments as described in detail below.

It appears that there is substantial divergence between the data and the model presented. The data in Figure 1 do not support the proposition that DENND6A localizes to lysosomes. A correlation coefficient of <0.1 would normally indicate that two proteins do not colocalize. While this is 'significantly' increased compared to GFP, one might expect this from any punctate or reticulate staining pattern in the juxtannuclear region of the cell. At best, the authors can state that DENND6A-GFP partially overlaps with LAMP1 staining. It is unclear whether the authors wish to claim that DENND6A localizes to lysosomes (see results section heading) or that distinct 'puncta and tubular structures of DENND6A were associated with LAMP1 vesicles' and 'wrapped around the LAMP1 vesicles.' To me, there is an important distinction between these two possibilities. The former is consistent with the model presented in Figure 9. The latter is not.

Building on this point, prior studies have indicated that Rab34 is a Golgi-localized protein (Wang and Hong (2002) 10.1091/mbc.e02-05-0280, Goldenberg et al. (2007) 10.1091/mbc.E06-11-0991, and Starling et al. EMBO Reports (2016) 10.15252/embr.201541382) and acts on lysosomes from the Golgi compartment (Wang and Hong (2002) 10.1091/mbc.e02-05-0280). The data presented in Figure 5F in the present study appear consistent with the above studies, and (in this reviewer's interpretation) the non-lysosomal localization of DENND6A presented here may also be consistent with this.

It is important that the authors adapt their manuscript and model to include consideration of this possibility or provide stronger evidence to refute it. This might, for example, include endogenous immunostaining of DENND6A clearly 'on lysosomes,' rather than associated with another compartment, with evidence that endogenous Rab34 is recruited to lysosomes in a DENND6A-dependent manner (I am not aware if Rab34 comes up in any lysosome proteomic studies) and that they localize together with RILP.

Response to reviewer 1:

We thank the reviewer for their comments. We wish to acknowledge that there was an oversight on our part in utilizing the correct parameter for reporting correlation coefficient values and for this, we sincerely apologize. We made an error in the process of thresholding and masking the region of interest (manually marked along the cell periphery to selectively calculate the correlation coefficient from within the cell while excluding the area outside). We have rectified this error in the revised manuscript and the updated correlation coefficient values now provide a more accurate estimation of DENND6A's partial presence on lysosomes (Fig. 1C and Fig. 7C of the revised manuscript). The updated correlation coefficient for DENND6A/LAMP1 is found to be ~ 0.31 . Furthermore, in response to the suggestion of reviewer 3, we have included new live-cell imaging data illustrating the dynamics of DENND6A with LysoTracker, providing better visualization of DENND6A's lysosomal localization (Supp Fig. 3 of the revised manuscript). Given the input from reviewer 1, we agree with the need for a more accurate representation of our findings. Even though our new data better supports the localization of a pool of DENND6A to lysosomes, we have revised the section heading on page 6 from results section to 'DENND6A partially localizes to lysosomes'.

In response to the comment of reviewer 1 regarding the limited lysosomal localization of DENND6A, which was also noted by reviewer 3 (comment 7), we performed new high-resolution imaging. Specifically, we conducted SIM imaging for DENND6A/Rab34/LAMP1 and DENND6A/Rab34/GM130 (Supp Fig. 6 in the revised manuscript). GM130 was chosen given the previous studies localizing Rab34 to the Golgi. While Rab34 aligns well with GM130, DENND6A exhibits minimal alignment with GM130. We thus hypothesize the existence of multiple GEFs for Rab34 that govern its localization to different organelles. For instance, as our understanding of the spectrum of substrates for DENN domain proteins expands, we have previously demonstrated that DENND1B activates Rab35 at the endosomal membrane (PMID: 20159556), while DENND2B activates Rab35 at the cytokinetic bridge (PMID: 37454296). Similarly, we propose that DENND6A may not be primarily responsible for Rab34's Golgi localization, as GFP-Rab34 retained its juxtannuclear localization even in DENND6A KO cells and was unable to cluster lysosomes (Fig. 5J in both the original and revised manuscript). We know that Rabs are generally found in a cytosolic state when they are inactive and bound to GDP-dissociation inhibitors. Inactive Rabs can be recruited directly to their respective membrane organelles once their GEF is recruited to the membrane (PMID: 16882731; PMID: 23382462). Accordingly, we propose in Figure 9 that DENND6A recruits inactive Rab34 from the cytosol to the lysosomes. Our data indicate that DENND6A relies on Rab34 to promote the juxtannuclear clustering of lysosomes, notably the Rab34 QL mutant rescues the lysosomal positioning defect observed in DENND6A KO cells (Figure 5J-K in the

revised manuscript). However, in response to the reviewer's suggestion to adapt our manuscript based on the previously reported localization of fluorescently tagged Rab34 at the Golgi, we have now added a discussion section about the possibility of potential crosstalk between lysosomes at the juxtannuclear site and the Golgi.

The reviewer suggests examining the localization of endogenous Rab34. There is a Rab34 antibody that has been employed for endogenous staining in several studies. While this antibody is effective for Western blot applications, we could not validate its specificity for immunofluorescence using a knockdown approach.

Regarding placement of Arl8b/DENND6A in figure 9, we postulate that DENND6A is recruited by Arl8b to peripheral lysosomes. Existing literature supports the presence of Arl8 on peripheral lysosomes. Our data, including the complete repositioning of Arl8b from the periphery to the juxtannuclear region upon co-expression with DENND6A and the reduction of the juxtannuclear pool of DENND6A in the absence of Arl8, convincingly indicate Arl8b's role in recruiting DENND6A to peripheral lysosomes and driving its relocalization to the juxtannuclear site. Unfortunately, analyzing the peripheral localization of DENND6A is challenging for two main reasons: 1) the lack of a validated antibody suitable for immunofluorescence to probe endogenous DENND6A, and 2) DENND6A overexpression leads to clustering of lysosomes near the juxtannuclear region, further complicating peripheral localization analysis.

Similarly, with the localization of Rab34 to the Golgi, it's challenging to explain how this Golgi localization alone could lead to the complete clustering of lysosomes, including those in the peripheral pool. We propose that cytosolic Rab34 plays a role by acting on peripheral lysosomes, contributing to the observed lysosomal clustering phenotype. However, the challenge in analyzing Rab34's association with peripheral lysosomes arises from the fact that the expression of Rab34 itself induces complete lysosomal clustering (PMID: 16473629 and in this manuscript).

We have incorporated all the above considerations into the discussion of the revised manuscript.

Reviewer #2:

In the manuscript by Kumar et al., the authors examine the role that DENND6A plays in lysosomal positioning via interaction with ARL8 and Rab34 activation, which the authors propose leads to retrograde trafficking of lysosomes from the periphery to the perinuclear region. This is an interesting paper that adds critical details to the mechanisms of lysosomal trafficking and assigns a role for DENND6A in this process. Additionally, the authors show that disruption of DENND6A results in reduced autophagic flux upon amino acid starvation, adding to pathway relevance. Regardless, there are a few points on DENND6A localization that require clarification, as well as some additional controls – as detailed below.

Response to reviewer 2: We thank the reviewer for their positive evaluation of our study. We have addressed the reviewer's comments as described in detail below.

Main Points:

1) Where is the majority of DENND6A localized? While the authors convincingly show that a small pool of DENND6A is at lysosomes, most of it is not. This makes it less clear as to what is happening to these other compartments upon manipulation of DENND6A levels. Is it possible that disruption of these other

compartments could be indirectly affecting lysosomal function and/or autophagy? Is DENND6A on earlier endosomal compartments?

Response to reviewer 2, comment 1:

We have performed new experiments examining for localization of DENND6A to early endosomes but observed no co-localization between DENND6A and early endosome marker (Supp Fig. 1B in the revised manuscript). As described in response to reviewer 1, we acknowledge an oversight related to the parameters for reporting correlation coefficient values for colocalization of DENND6A and LAMP1, and for this, we sincerely apologize. We made an error in the process of thresholding and masking the region of interest (manually marked along the cell periphery to selectively calculate the correlation coefficient from within the cell while excluding the area outside). This error has been corrected in the revised manuscript (Fig. 1C and Fig. 7C of the revised manuscript). This demonstrates that DENND6A/LAMP1 has a correlation coefficient of ~ 0.31 , indicating localization of DENND6A to lysosomes. DENND6A has been reported to localize on recycling endosomes in conjunction with Rab14 (PMID: 22595670), a topic that we discussed in the original manuscript's discussion section and now also mention in the results section.

As mentioned in the original manuscript, our work has revealed associations between DENND6A and several Rab proteins, with each Rab potentially having a distinct localization. While the functions of DENND6A in conjunction with these other Rab partners remain to be elucidated, we do not think that disruptions in other compartments indirectly affect lysosomal function. This is substantiated by the observation that Rab34 phenocopies DENND6A in terms of lysosome positioning, and the rescue of the DENND6A knockout phenotype by Rab34QL. If DENND6A/Rab14 on recycling endosomes were indirectly involved, we should not see such a robust rescue with Rab34QL.

Despite our experimental evidence demonstrating a pool of DENND6A on lysosomes, we acknowledge the reviewer's comment that we cannot definitively rule out the possibility of crosstalk with other compartments. Therefore, we have included in the discussion the notion that given the broad spectrum of potential GTPase targets for DENND6A, it remains plausible that interactions with other compartments under different physiological conditions, each harboring DENND6A and other Rab partners, may exist. These interactions are likely to become more apparent as future studies on these other pairs are reported.

Additionally, as suggested by reviewer 1, we have made modifications to our text saying "DENND6A partially colocalizes with lysosomes."

2) Related to the above, the authors use exogenous GFP-DENND6A for localisation, which could be mislocalized – is it possible to visualize endogenous DENND6A? What is the expression of GFP-DENND6A relative to endogenous DENND6A?

Response to reviewer 2, comment 2: In the revised manuscript, we have included new experiments to include both N-terminal and C-terminal GFP-tagged DENND6A (Fig. 1B-C and Supp Fig. 1A in the revised manuscript), demonstrating that both exhibit a similar expression pattern and that both drive lysosomal clustering (Fig. 2C in the revised manuscript). Unfortunately, we lack an antibody suitable for immunofluorescence (IF) to visualize the endogenous localization of DENND6A. We now include quantification for the expression levels of DENND6A-GFP relative to endogenous DENND6A (Supp Fig. 2 in the revised manuscript). We find exogenous DENND6A to be 1.8-times higher than endogenous. At this level, we do not believe that there should be overexpression artifacts.

3) In Figure 7F, loss of ARL8a/b results in a peripheral relocalization of DENND6A, but not of LAMP1 (as detailed in insets) – I was slightly confused here as to why lysosomes were not more peripheral too (as they cannot now recruit DENND6A), so maybe more explanation is needed? Likewise, as mentioned in point 1 – what is the identity of the DENND6A peripheral compartment here?

Response to reviewer 2, comment 3: We thank the reviewer for pointing out this inconsistency and apologize for the confusion. Arl8 is a major endolysosomal GTPase known to interact with kinesin and promote anterograde lysosomal transport. Loss of Arl8 has been reported to lead to the juxtannuclear clustering of lysosomes (e.g., PMID: 22172677), as mentioned by reviewer 3 in their “minor points-comment 3”. In the end, the image we selected was not truly representative, and for this we apologize. In the revised manuscript, we have chosen an image that better represents our quantification and aligns with the literature, showing clustered lysosomes (LAMP1) in the absence of Arl8. Additionally, we have included new quantification data in the revised manuscript, showing reduced colocalization of DENND6A with LAMP1 in the absence of Arl8 (Supp Fig. 12 in the revised manuscript).

In response to the query about the peripheral relocalization of DENND6A, it may have appeared more peripheral in the absence of its juxtannuclear pool. However, these compartments were distributed throughout the cytoplasm with some accumulation at the periphery. We have consistently observed peripheral and cytoplasmic DENND6A compartments/tubular structures. Notably, these peripheral and cytoplasmic DENND6A structures overlap with the transferrin receptor, a marker for recycling endosomes, suggesting the presence of a pool of DENND6A on recycling endosomes (as shown in Fig 3D from a published manuscript, PMID: 22595670; DENND6A is also known as FAM116A). The authors of this publication (PMID: 22595670) did not report DENND6A’s lysosomal localization, possibly because lysosomes were not the focus of their study.

4) The autophagy data in Figure 8 does indeed suggest reduced levels of autophagy in the absence of DENND6A, but it was not clear why total levels of LC3B were lower in these cells – this does not fit with the authors argument that the defect is at the level of autophagosome-lysosome fusion (if fusion was blocked then there should be more LC3-II, as with bafilomycin). The authors should blot for another marker such as p62/SQSTM1, which also accumulates when autophagy is blocked.

Response to reviewer 2, comment 4: The blot in Figure 8C does not represent total LC3B levels. The antibody used here is a KO-validated LC3B antibody that strongly reacts with LC3B-II, the levels of which are normalized to the loading control for quantification. Despite high-exposure blots, we could not detect LC3B-I. The LC3B-II band, were also observed in independent studies conducted by other research groups using the same antibody (references: PMID: 35314681, Fig. 8F; PMID: 33707434, Supp fig 2A-C; PMID: 34074205, Fig. 1B).

We apologize for any misunderstanding that may have arisen, leading the reviewer to believe that we implied a defect in autophagosome-lysosome fusion, suggesting a blockage in this process. The reviewer correctly points out that if fusion were blocked, we would expect to see increased LC3B-II levels. However, even under normal/unstarved conditions, we consistently observed reduced levels of LC3B-II. In our original manuscript, we proposed that the loss of DENND6A diminishes autophagy, potentially by impeding autophagosome formation.

To understand the underlying reasons for the observed decrease in the levels of lipidated LC3B (LC3B-II) in normal/unstarved conditions, in the original manuscript, we conducted 'starvation' and 'starvation +

bafilomycin' treatments. The decline in LC3B-II levels could result from two scenarios: reduced autophagosome formation, leading to decreased autophagy (resulting in lower levels of LC3B-II), or increased fusion between autophagosomes and lysosomes, leading to enhanced autophagic degradation and, once again, lower LC3B-II levels. Starvation typically induces autophagy by increasing autophagosome synthesis to provide essential nutrients. However, even when we induced autophagy through nutrient starvation, we consistently observed significantly reduced levels of LC3B-II in DENND6A KO cells (Fig. 8C-D in the original manuscript). To investigate whether the decreased LC3B-II levels in DENND6A KO cells were due to increased autophagosome-lysosome fusion, we treated starved cells with the inhibitor Bafilomycin, which blocks autophagosome-lysosome fusion and degradation. Despite this inhibition of fusion, we continued to observe lower levels of LC3B-II (Fig. 8C-D in the original manuscript), providing further evidence that the defect lies in autophagosome formation, resulting in reduced autophagy. We have modified the text in the result section to reflect on this explanation.

As per the reviewer's suggestion to include p62 levels, we have now incorporated p62 blots into the revised manuscript (Fig. 8C and Supp Fig. 14 in the revised manuscript). We consistently observe lower p62 levels in the DENND6A KO cells, suggesting reduced autophagy in the absence of DENND6A.

5) Is the autophagy block related to the defect in lysosomal positioning? It could be beneficial to expand the IF data shown in Figure 8E to include the EBSS conditions +/- Baf: are autophagosomes also more peripheral in absence of DENND6A or is it just the lysosomes?

Response to reviewer 2, comment 5: Yes, we think that the inhibition of autophagy is indeed linked to defects in lysosomal positioning. As mentioned in comment 6 by this reviewer, there is a possibility of defective mTOR signaling. It is known from previous literature that lysosome positioning plays a crucial role in coordinating mTORC1 activity, thereby regulating autophagosome synthesis and autophagosome-lysosome fusion (PMID: 21394080). Specifically, mTORC1 phosphorylates ULK1 on Ser 757 (in mouse; equivalent to Ser758 of human ULK1) to inhibit the interaction of ULK1 with and activation by AMPK, subsequently preventing autophagy (PMID: 21258367). However, during starvation, mTORC1 is inhibited, Ser757 phosphorylation of ULK1 is reduced, allowing ULK1 to interact with AMPK and initiate autophagy (PMID: 21258367). In line with the point addressed in 'Response to reviewer 2, comment 6,' our findings indicate that ULK1 Ser757 phosphorylation remained elevated even during starvation, supporting the notion of a block in autophagy initiation.

We have expanded the IF data shown in Figure 8E and now include EBSS +/-Baf A1 in the revised manuscript, which demonstrates consistent reduction in LC3B staining in EBSS +/-Baf A1 conditions.

We have included quantification of LC3B distribution in WT versus DENND6A KO cells (Supp Fig. 13 in the revised manuscript) to address autophagosome distribution. It appears that the autophagosomes are also peripheral in DENND6A KO cells, although not to the same extent as we observed with lysosomal distribution.

6) Related, maybe autophagy initiation is also blocked in absence of DENND6A? They could look at WIPI2 puncta formation or ULK1 substrate phosphorylation. This may relate to defect in mTOR signalling, especially given the changes in lysosome localization.

Response to reviewer 2, comment 6: We thank the reviewer for suggesting this experiment. We have now investigated ULK1 phosphorylation (Supp Fig. 15 of the revised manuscript), and our findings indeed indicate that while ULK1 undergoes dephosphorylation (Ser758) under nutrient starvation in WT cells,

the levels of phospho-ULK1 were significantly higher in DENND6A KO cells during starvation. This suggests that autophagy initiation is also impaired in the absence of DENND6A. Exploring this pathway more comprehensively in future studies will provide further insights into the defective mechanism when DENND6A is absent.

7) Finally, the authors could do with some extra controls. Given the phenotypes in the KO clones, can these be rescued by re-expressing WT DENND6A? Also, what happens if another DENND protein is overexpressed – does this result in a similar perinuclear clustering phenotype, or is it exclusive to DENND6A?

Response to reviewer 2, comment 7: We appreciate the reviewer's suggestion for a rescue experiment, also proposed by reviewer 3, comment 2. We have revised Figure 2E and Supp Fig. 4, which now illustrate the rescue of the lysosomal positioning phenotype by expressing DENND6A-GFP in the DENND6A KO cells.

Furthermore, in response to the reviewer's recommendation, we have introduced an additional control by expressing another DENN domain protein, DENND2B, which does not rescue the dispersed lysosomal phenotype in DENND6A KO cells (Figure 2E and Supp Fig. 4 in the revised manuscript).

Reviewer #3:

In the present work, Kumar et al. have investigated the role of DENN domain protein family member DENND6A in retrograde transport and positioning of LAMP1-positive compartments, collectively referred to as lysosomes. The authors show that DENND6A partially localizes to lysosomes, and its overexpression results in juxtannuclear positioning of lysosomes. Conversely, depletion of DENND6A leads to the peripheral distribution of lysosomes. Further, using the FRB-FKBP-rapamycin system, the authors show that DENND6A is sufficient to cause clustering of target organelles. DENND6A is a member of the DENN domain family of proteins, which constitute the largest family of Rab GEFs. The authors have used a relocalization approach (targeting of DENND6A to mitochondria) to identify potential Rab binding partners for DENND6A. The screening resulted in the identification of 20 Rabs, from which the authors have further characterized the interaction of DENND6A with Rab34, a golgi- and endolysosomal-localized Rab GTPase. The author shows that DENND6A exhibits GEF activity towards Rab34, and Rab34 is required for DENND6A-mediated lysosome positioning. Previously, Rab34 was shown to mediate juxtannuclear clustering of lysosomes and required interaction with its effector, RILP, a dynein adaptor. Using a recombinant protein pulldown approach, the authors show that DENND6A interacts with the Rab34-RILP-DIC (the intermediate chain of the dynein motor machinery) complex.

Previously, using a BioID-based interactome, DENND6A was reported as a potential binding partner of the lysosome-localized small GTPase Arl8b. In the present manuscript, the authors confirmed this interaction and showed that DENND6A acts as an effector of Arl8b and colocalizes with Arl8b in LAMP1-positive compartments. Arl8b regulates the positioning of lysosomes in cells by recruiting specific effectors for the loading of dynein or kinesin motor proteins on lysosomes. Interestingly, the authors report that silencing of Arl8a/b alters the perinuclear distribution of DENND6A-compartments. Finally, the authors have shown that DENND6A regulates the nutrient-dependent positioning of lysosomes and positively regulates autophagic flux.

Overall, this is an interesting study, as the authors have characterized the role of DENND6A in lysosome positioning, which was previously unknown. However, the study in its present form falls short of the impact. Particularly, some of the major conclusions drawn by the authors need more experimental evidence (with the addition of key controls and quantification), which is currently missing in the manuscript. Below are specific concerns about some of the experiments and their presentation.

Response to reviewer 3: We thank the reviewer for their overall positive evaluation of our study. We have addressed the issues mentioned above as described in detail below.

Major concerns:

1) In Fig. 1B-E, colocalization of overexpressed DENND6A-GFP with LAMP1-positive compartments is quite low (PCC value of ~ 0.15). Representative images also show DENND6A-GFP localization to tubular and non-LAMP1-positive structures. First, the authors should reassess the colocalization by including more cell numbers from at least three independent experiments, as the data plotted is only from 25 cells, as per the figure legend. The identity of non-LAMP1-positive DENND6A compartments should be established by the authors. Also, authors should confirm whether the C-terminal GFP-tagging of DENND6a is not causing mislocalization, and therefore, N-terminal GFP-tagging or the use of small tags such as FLAG and HA should be explored. If possible, the authors should try to visualize the endogenous distribution of DENND6A in HeLa cells.

Response to reviewer 3, comment 1: We would like to thank the reviewer for their valuable feedback. Regarding our colocalization studies: As highlighted in our responses to reviewers 1 and 2, we recognize that an oversight occurred in our use of the correct parameter for reporting correlation coefficient values, and for this, we sincerely apologize. We made an error in the process of thresholding and masking the region of interest (manually marked along the cell periphery to selectively calculate the correlation coefficient from within the cell while excluding the area outside). We have rectified this error in the revised manuscript and the updated correlation coefficient values now provide a more precise estimation of the presence of a pool of DENND6A on lysosomes (Fig. 1C and Fig. 7C of the revised manuscript). Additionally, the initial count of 25 cells in total was derived from three independent experiments, with 8-9 cells per repeat. In the revised manuscript, we reevaluated a total of 30 cells from three independent experiments, which led to significantly improved colocalization values (Fig. 1C of the revised manuscript).

As mentioned in our response to reviewer 2, one of the other non-LAMP1-positive DENND6A compartments overlaps with a population of the transferrin receptor, a recycling endosome marker (as shown in Fig 3D from PMID: 22595670). At the suggestion of reviewer 2, we performed early endosome staining (Fig. 1C and Supp Fig. 1B in the revised manuscript), and we did not detect any colocalization with DENND6A. In response to reviewer 3's suggestion (comment 8), we performed SIM imaging for Golgi. Although we found mScarlet-Rab34 to be predominantly localized to the Golgi, consistent with previous literature, DENND6A did not exhibit Golgi localization (Supp Fig. 6 in the revised manuscript). Finally, we currently lack a suitable DENND6A antibody for conducting immunofluorescence studies to examine the endogenous localization of the protein.

To ensure that C-terminal GFP tagging of DENND6A is not inducing mislocalization, we have now expressed GFP-DENND6A, which exhibits the same expression pattern – a juxtannuclear pool with cytoplasmic compartments (Fig. 1B-C and Supp Fig. 1A in the revised manuscript), and it drives lysosomal clustering (Fig. 2C in the revised manuscript).

2) The authors should confirm that DENND6A-GFP is a functional protein and can rescue the effects of loss of DENND6A on lysosome positioning as shown in Figs. 2E–F. Also, the effect of lysosome positioning should be rescued with a GEF activity-deficient mutant of DENND6A if the authors need to establish that DENND6A GEF activity is required for lysosome positioning.

Response to reviewer 3, comment 2: We have revised Figure 2E and Supp Fig. 4, which now demonstrate the rescue of the lysosomal positioning phenotype by expressing DENND6A-GFP in the DENND6A KO cells.

In response to using a GEF activity deficient mutant of DENND6A, we considered this experiment carefully but encountered challenges. There are seven residues essential for catalytic GEF activity toward Rab GTPases based on the structure of the DENN domain of DENND1B (PMID: 22065758). We have successfully generated GEF deficient DENN domain family members in previous studies (PMID: 33443570), including one in DENND2B (as shown in Figure 1 of the response letter). Regrettably, we could not find an alignment between DENND1B and DENND6A allowing us to determine residues critical for GEF activity (Figure 1 of the response letter).

Query: DENND1B / Subject: DENND1A						Query: DENND1B / Subject: DENND2B					
Score	Expect	Method	Identities	Positives	Gaps	Score	Expect	Method	Identities	Positives	Gaps
568 bits(1464)	0.0	Compositional matrix adjust.	276/425(65%)	328/425(77%)	24/425(5%)	137 bits(346)	4e-39	Compositional matrix adjust.	110/394(28%)	191/394(48%)	54/394(13%)
Query 1	MDCRTKANPDRFTDLVLKVKCHASEN--	EDPVVLWKFPEDFDQEQILOSVPKFCFPDVE	58			Query 42	QEILQSVPKFCFPDVERVSQNG-VGQHTFVLTDIESKQRFGFCR--	LTSGGTICL---	95		
Sbjct 1	M R K NP+ TF++ ++V + DP V +FPED+ DQE+LQ++ KFCFPF V+ MGRSRIKQNPETTFFVVEVAYVPRGGT	LSDEPVRQRFPEQYSQEVLTLTKFCFPFVVD	60			Sbjct 736	EERLKAIQFCFPDAKDMLVPVSEYSSETFSFMLTGEDGSRFFGYCRRLLPSGKGPRLP	LEV	795		
Query 59	RVSQNQVQGHFTFVLTDIESKQRFGFCRLTSGGTICL	CILSYLPWFVEVYKLLNLADYL	118			Query 96	-CILSYLPWFVEVYKLLNLADYLAKELENDLNETL----	RSLYHPVKANTPVNLSV	149		
Sbjct 61	++ +QVQ+FTFVLTDI+SKORFGFCRL+SG C CILSYLPWFVEV+YKLLN LADY SLTVS	QVGNFTFVLTDI+SKORFGFCRLSSGAKSCFCILSYLPWFVEVYKLLNLADYT	120			Sbjct 796	YCVISRLGCFGLFSKVLDEV-----	ERRRGISAALVYPMRSLMESPPAPGK--TIKV	847		
Query 119	AKELNDLNETLRLSYHPVKANTPVNL	SNVQEIFIACEQVLKDQPALVPHSYFIAPDV	178			Query 150	NQEIFIAEQVLKDQPALVPHSYFIAPDV	TGLTIPESRNLTEYFVAVDNNMLQYAS	209		
Sbjct 121	K EN NE L +L+ P+P V+LSV HSYF PD TKRQENQWNLLETLHKLPI	DPGVSVHLSV-----HSYFTVDPD	160			Sbjct 848	KTFLPAGNEVLE-----	LRRPMSRL----EHVDFECLFTCLSVRQLIRIFASL	893		
Query 179	TGLPTIPESRNLTEYFVAVDNNMLQYASMLHERRI	VISSKLSLTACIHGSAALLYP	238			Query 210	LHERRIVISSKLSLTACIHGSAALLYP	YQHTIHPVLPPIHCCAPFYLIGIHS	269		
Sbjct 161	LP+IPE+RNLTEYFVAVDNNML LYASML+ERRI+II SKLSTL	TACIHGSA+LYP RELPSIPENRNLTEYFVAVDNNML	LYASMLYERRI	LIICSKLSLTACIHGSAAMLYP	220	Sbjct 894	LLERRIVFVADKLSLSSSHAWALLYP	FSQHHTFVLPASMTDICCPTFLVGLLS	953		
Query 239	MYWQHNPVLPPIHCCAPFYLIGIHS	SLIERVKNKSLIEDVWMLNVDNTL	ESPF			Query 270	SLIERVKNKSLIEDVWMLNVDNTL	ESPFDLNN-LPSDVVSALKNKLKKQST-----	320		
Sbjct 221	MYWQHNPVLPPIHCCAPFYLIGIHS	SLIERVKNKSLIEDVWMLNVDNTL	ESPF			Sbjct 954	SSLPKLKELVPEEALMVNLGSDR	FIQMDDEDTLPRKLAQALEALERKNELISQSDS	1013		
Query 299	DLNLPDSVVSAKLNKLNKQSTATGDGVARAFL	RAQAALFGSYRDALRYKGPETIFCEE	358			Query 321	-----ATGDG-VARAFL	RAQAALFGSYRDALRYKGPETIFCEE	372		
Sbjct 281	DL +LP+DV+S+LKN+LKK ST TGDGVARAFL+AQAA FGSYR+AL+ +P EPIITFCEE DLQSLPNDVISLKNRLLKVS	TTTGDGVARAFLKAQAALFGSYRNALKIEPEEPIITFCEE	340			Sbjct 1014	DSDDCENTLNLVSEV	IRFFVETVGHYSLFTQSEKGERAFQREAFRKSVAKSIRRF	1073		
Query 359	SFVKH-RSSVMKQFL	EATAINLQFLFKQFIDGRLAKLNAGRGFSDFVEEII	SGGFCGGDK			Query 373	EATAINLQFLFKQFIDGRLAKLNAGRGFSDFVEEII	406			
Sbjct 341	+FV H RS M+QFL+ A LQFLFKQFIDGRL LN+G GFSDFVEEII G + G DK AFVSHYRSGAMRQFLQATQ	LQFLFKQFIDGRLDLLNSGEGFSDFVEEII	INMGVEA-GSDK			Sbjct 1074	EVFMESSMFAGIQDRELKCRAG--LFEQRV	1104			
Query 418	LQYDY 422										
Sbjct 400	L + + LYHQW 404										

Query: DENND1B / Subject: DENND6A -  No significant similarity found. For reasons why, click here

Figure 1

AlphaFold structures revealed major structural deviations when comparing DENND6A to the solved DENND1B structure:

1) A portion of the 'conserved' catalytic residues are situated within the C-terminal helical tail of the DENN domain from DENND1B. However, AlphaFold structure predictions indicate that the C-terminus of DENND6A has a different structural arrangement. In DENND1B, the DENN domain ends with a short helical tail, but this helix is greatly extended in DENND6A and folds inward on itself, creating a unique tertiary structure not observed in DENND1B.

2) Furthermore, after structurally aligning DENND6A with DENND1B, the alignment at other regions of interest was rather poor. The other 'conserved' residues lie in regions where there are substantial differences in secondary structure, making direct comparisons between the two difficult.

Despite these challenges, we have presented four lines of evidence supporting DENND6A as a GEF for Rab34:

- 1) Mito-recruitment assay.
- 2) Binding experiment demonstrating a preferential interaction with the inactive mutant, a characteristic of a GEF/substrate relationship.
- 3) Effector binding assay.
- 4) *In vitro* GEF assay.

3) Does DENND6A-GFP localize to acidic and degradative lysosomes as well? This is important to determine as both the non-degradative and degradative populations will be marked by LAMP1. Preferably, authors should do these experiments in live cells using dyes such as LysoTracker and/or SiR-lysosome or Magic Red. These dyes generally show diffuse localization in fixed cells.

Response to reviewer 3, comment 3: We thank the reviewer for suggesting this experiment. We have added this new data in Supp Fig. 3 and Movie 3-4 of the revised manuscript. Live cell imaging of DENND6A/LysoTracker revealed a better visualization of the dynamic association of DENND6A with acidic compartments marked by LysoTracker.

4) In the context of the above point, the authors should report the effect of DENND6A overexpression on Rab7-positive compartments. Does DENND6A specifically reposition Arl8b-positive endolysosomes and not Rab7-marked endolysosomes? This will be important as RILP is a well-known effector of Rab7.

Response to reviewer 3, comment 4: We acknowledge the significance of this question, but it deserves a comprehensive investigation due to its complexity. Several factors contribute to the complexity. A subset of late endosomes/lysosomes contains both Arl8b and Rab7 (PMID: 28325809; PMID: 32080880), making it challenging to analyze this aspect. Rab7 also serves as a late endosome/lysosome marker. The expression of DENND6A alone effectively clusters lysosomes at the juxtannuclear site, which further complicates the specific analysis of repositioning Rab7-marked compartments. Additionally, although RILP is shared by Rab7, Rab7 was not identified in our DENND6A screen, and therefore we did not pursue this aspect further.

5) As shown in Fig. 3, DENND6A targeted to peroxisomes is sufficient to cause their perinuclear clustering. As in the author's model (Fig. 9), DENND6A does not directly bind the dynein motor but via the RAB34-RILP complex. The authors should test whether DENND6A-mediated peroxisomal clustering is Rab34- and RILP-dependent.

Response to reviewer 3, comment 5: We have added new data in Supp Fig. 9 of the revised manuscript demonstrating that the ability of DENND6A to induce peroxisome clustering depends on RILP and dynein.

6) In Fig. 5C, authors should include a control panel showing levels of another endolysosomal Rab protein (such as Rab7 since it was not recruited by DENND6A as per the screening data mentioned in Fig. 4) to strengthen the conclusion that overexpression of DENND6A increases the active levels of Rab34 in the cell. Similarly, the inclusion of a GEF activity-defective mutant of DENND6A in Fig. 5E should be tested.

Response to reviewer 3, comment 6: Following the reviewer's suggestion, we conducted a Rab7 blot, and observed elevated Rab7 levels in DENND6A-expressing cells (Figure 2 of the response letter). We were not surprised because this finding aligns with a study that revealed the formation of a complex between active Rab34 and active Rab7 in the presence of RILP (PMID: 27113757, Fig 6I). In this earlier study, Rab34 and Rab7 were labeled differently (GFP-Rab34 and HA-Rab7), and immunoprecipitation using GFP brought together RILP and Rab7. However, the functional significance of this complex remains unclear due to the lack of experimental data. It appears in our studies that DENND6A-mediated increases in the levels of endogenous Rab34 may lead to more pull out of endogenous Rab7.

We have opted not to include this new data in the revised manuscript as it is likely to divert focus from the main topic. In relation to these findings, we also do not think that Rab7 is involved in DENND6A/Rab34-mediated lysosomal clustering because 1) DENND6A does not recruit Rab7 in our screen, and 2) the expression of Rab7 alone failed to induce juxtannuclear clustering of lysosomes (Fig 3A from PMID: 27113757).

Figure 2

Regarding the use of a GEF activity-defective mutant of DENND6A, we have provided detailed explanations in 'Response to reviewer 3, comment 2' outlining the reasons why we were unable to conduct this experiment.

7) Based on the results shown in Figs. 5J–K, the authors have concluded that Rab34 is unable to mediate perinuclear positioning of lysosomes in the absence of DENND6A. However, from the images shown, it seems Rab34 is still able to mediate the perinuclear positioning of LAMP1-positive compartments in DENND6A KO cells (compare the positioning of LAMP1-positive compartments in untransfected vs.

Rab34-transfected cells in DENND6A KO HeLa cells). Can the authors clarify this point? To establish that Rab34 is downstream of DENND6A-mediated GEF activity, authors can express a constitutively active version of Rab34 and check whether that rescues the effect of DENND6A KO on lysosome positioning.

Response to reviewer 3, comment 7: We appreciate the reviewer's feedback and respectfully acknowledge a differing interpretation. Our conclusion regarding the inability of Rab34 to induce juxtannuclear clustering in DENND6A KO cells can be better understood by comparing the clustering observed in WT+Rab34 and DENND6A KO+Rab34 conditions. In WT+Rab34, we observe complete clustering, whereas in DENND6A KO+Rab34, lysosomes maintain a peripheral distribution, even in the two cells where minor clustering was noted by the reviewer. This variation is likely due to inherent cell-to-cell differences within a population that still exhibit significant peripheral lysosomes and a minor juxtannuclear pool. As illustrated in Figure 2E and 8A (under unstarved conditions), both in the revised and original manuscripts, all KO cells display a substantial peripheral fraction of lysosomes, and some of them also exhibit a minor juxtannuclear pool. The reviewer's perception could have been affected by the small size of the figure. To address this issue, we have included an enlarged view of the main Fig. 5J-K (Figure 3 in the response letter):

Figure 3

As per the reviewer's suggestion, we have also included the rescue experiment data by expressing the constitutively active mutant of Rab34 in DENND6A KO cells (Fig. 5J-K in the revised manuscript).

8) In Figs. 5F and 5J, Rab34 localization looks very perinuclear and resembles Golgi staining. Have the authors checked Rab34 and DENND6A-GFP colocalization with Golgi? The image in Fig. 5F should be again performed with SIM imaging, as it is not clear whether mCherry-Rab34 and DENND6A-GFP are colocalized with LAMP1. Also, authors should analyze the localization of Rab34 (endogenous if possible) on lysosomes in DENND6A KO cells.

Response to reviewer 3, comment 8: As reviewer 1 highlighted, fluorescently tagged Rab34 is primarily localized to the Golgi. In response to the suggestion by reviewer 3, we conducted additional SIM imaging for DENND6A/Rab34/LAMP1 and DENND6A/Rab34/GM130 (Supp Fig. 6 in the revised manuscript). Our results reveal that while Rab34 overlaps with GM130, DENND6A shows minimal overlap with GM130, with very few contact sites. However, in the case of LAMP1, we have identified colocalized regions of DENND6A/Rab34/LAMP1.

Regarding the use of the Rab34 antibody for endogenous localization, we appreciate the reviewer's comment. While this antibody has been employed for endogenous staining in previous studies, we cannot validate its specificity for immunofluorescence using a knockdown approach.

9) The authors report an exciting result wherein DENND6A is shown to act as an effector of Arl8b. However, this was established based on a GST-pulldown assay (Fig. 7A). The authors should strengthen this data by performing additional assays. Does Arl8b interact with DENND6A in cells in a GTP-dependent manner? This can be tested by performing co-immunoprecipitation experiments using GTP/GDP-locked mutants of Arl8b. If DENND6A is an Arl8b effector, then mito-Arl8b (mitochondrial localized Arl8b) should recruit it on the membrane. Further, DENND6A localization to lysosomes should be reduced in Arl8b-silenced cells. This could be tested by performing confocal imaging and biochemical analysis using the Lyso-IP approach. In the same experiments, levels of Rab34 on lysosomes can be determined.

Response to reviewer 3, comment 9: We have conducted co-immunoprecipitation experiments, as suggested by the reviewer, indicating that DENND6A preferentially interacts with the GTP-locked mutant of Arl8b (Supp Fig. 10 of the revised manuscript).

Additionally, in response to the reviewer's suggestion, we attempted to target the inactive and active forms of Arl8b to mitochondria to investigate the preferential recruitment of DENND6A (Supp Fig. 11 of the revised manuscript). We did not observe any recruitment in this context. We speculate that the lipid composition of the lysosomal membrane may also play a role in the recruitment of DENND6A, a possibility that warrants further exploration. A similar phenomenon has been observed in the recruitment of the Arf GTPase effectors, four-phosphate-adaptor protein 1 and 2 (FAPP1 and FAPP2), which interact with PtdIns(4)P and Arf through their pleckstrin homology domain for localization to the trans-Golgi network (PMID: 15107860). This additional information has been incorporated into the result section of the revised manuscript.

Reviewer 3 raised the valid point that the localization of DENND6A to lysosomes should be reduced in Arl8b-silenced cells. Regarding biochemical analysis, it is noteworthy that, as emphasized by all the reviewers, including this reviewer, the proportion of DENND6A associated with lysosomes is indeed relatively modest, a fact also evident from the Lyso-IP experiment (Fig. 1F), wherein the degree of DENND6A enrichment is lower compared to that of LAMP1. Additionally, as suggested by reviewer 1, we have updated the heading in the results section of the revised manuscript to '...DENND6A partially localizes to lysosomes.' Given the limited pool of DENND6A on lysosomes, detecting its reduced presence on lysosomes is technically challenging using the Lyso-IP method. However, we have addressed the reviewer's concern by including quantification of confocal images demonstrating reduced colocalization of DENND6A with lysosomes in control cells as compared to the Arl8 siRNA-treated cells (Fig. 7F and Supp Fig. 12 of the revised manuscript).

10) The authors should test if DENND6A competes with SKIP for binding to Arl8b in order to reposition lysosomes towards the cell center. Recently, RUFY3 was shown to act as an effector of Arl8b to mediate the perinuclear positioning of lysosomes. How are these effectors of Arl8b (RUFY3 and DENND6A) mediating juxtannuclear repositioning of lysosomes? Is RUFY3 also present in complex with DENND6A? Is RUFY3 still able to relocalize Arl8b-positive lysosomes in DENND6A KO cells?

Response to reviewer 3, comment 10: We agree with the reviewer that DENND6A may compete with SKIP/kinesins for binding to Arl8b, and we have now included this caveat in the discussion of the revised manuscript. This is a significant question and addressing it would necessitate a detailed study to determine the nature of competition among these effectors for binding to Arl8b.

The loss of RUFY3 does not affect autophagic flux, whereas the loss of DENND6A does, despite both being effectors of Arl8b. Therefore, we do not think that RUFY3 is present in a complex with DENND6A. Our hypothesis is that there may be a subpopulation within lysosomes where these effectors are recruited in the presence of Arl8b under specific physiological conditions. This again underscores the critical question of competition among these effectors, which requires further investigation in a comprehensive study.

11) As per the authors' model (Fig. 9), Arl8b recruits DENND6A, which in turn recruits Rab34 on lysosomes. This is not shown in the study. Does DENND6A directly interact with Arl8b? Identification of the Arl8b-binding site on DENND6A and further rescue with an Arl8b-binding defective mutant of DENND6A will further strengthen the claim that Arl8b recruits DENND6A to mediate Rab34 GDP-GTP exchange on lysosomes.

Response to reviewer 3, comment 11: As detailed in "Response to reviewer 3, comment 9," we hypothesize that, besides Arl8b, an unidentified factor, which could potentially be linked to the lipid composition of the lysosomal membrane, is essential for recruitment of DENND6A to lysosomes. Conducting the experiment suggested by the reviewer is challenging without knowledge of this other factor. Given the indications of the involvement of this unknown factor, we have updated Figure 9 and the discussion in the revised manuscript to convey the possible presence of an additional component influencing DENND6A recruitment.

12) It is surprising that the authors have not explored the role of DENND6A-mediated lysosome positioning on lysosome fusion with late endosomes and thereby cargo degradation. These experiments will signify the impact of DENND6A-mediated positioning of lysosomes on their function in cargo degradation.

Response to reviewer 3, comment 12: Lysosomal positioning plays an important role in controlling myriad of cellular processes (PMID: 27799357). Our primary focus has been on autophagy, as prior research has indicated that RUFY3-mediated lysosomal positioning does not affect autophagy. We acknowledge the valuable question raised by the reviewer and we were planning to study this aspect in detail in the future. Because each of these numerous cellular processes including endocytic and phagocytic degradation requires its own set of detailed experiments, it goes beyond the scope of one manuscript.

However, in response to the reviewer's inquiry, we have incorporated a new data (Supp Fig. 16 in the revised manuscript) to address this aspect. Specifically, we conducted an assay using DQ-BSA, a method recommended in previous literature (PMID: 16982798; PMID: 28325809), to examine endocytic

cargo trafficking to lysosomes and its subsequent degradation. DQ-BSA is an endocytic cargo that fluoresces upon proteolytic cleavage within lysosomes. Our findings indicate that the loss of DENND6A also impairs endocytic cargo degradation. We have addressed this result in the discussion section of the revised manuscript and emphasized the necessity for additional experiments to accurately identify the dysfunction within this pathway.

Minor points:

1) If possible, the authors should test the effect of Arl8b overexpression or knockdown on Rab34 and DENND6A interactions.

Response to reviewer 3, minor comment 1: DENND6A/Rab34 functions downstream of Arl8b, with Arl8b primarily serving to recruit DENND6A as one of its effectors. Consequently, we believe that Arl8b does not influence DENND6A's functionality toward Rab34. This perspective is also reinforced by the fact that the in-vitro GEF assay does not necessitate the presence of Arl8b, yet the exchange reaction still occurs. As a result, we did not investigate the impact of Arl8b on the Rab34/DENND6A interaction.

2) In Fig. 6A, using a recombinant RILP protein in a pulldown assay, the authors found the presence of a complex containing DENND6A, Rab34, dynein, and RILP, suggesting that they function together in the same pathway. The authors should test the presence of this complex in vivo by co-immunoprecipitation and, if possible, probe for the presence of dynactin subunits in the complex.

Response to reviewer 3, minor comment 2: As per the reviewer's suggestion, we performed the co-immunoprecipitation (co-IP) experiment. The DENND6A antibody was not suitable for IP. Therefore, we expressed GFP or DENND6A-GFP in HEK-293 cells and performed GFP IP, demonstrating that the complex comprises of Rab34, dynein, RILP and p150 (a dynactin subunit) (Supp Fig. 8 of the revised manuscript).

3) The results shown in Fig. 7F on LAMP1 positioning in Arl8a/b siRNA are somewhat confusing. Several studies have reported that Arl8a/b silencing leads to perinuclear positioning of lysosomes in HeLa cells. However, the LAMP1 staining shown in control vs. Arl8a/b siRNA seems to be the same. Can the authors clarify this point?

Response to reviewer 3, minor comment 3: We thank the reviewer for highlighting this inconsistency. The discrepancy arose from our selection of representative image. Therefore, we revisited the images and performed a careful analysis. In the revised manuscript, we have replaced the image with one that better aligns with the literature, showing clustered lysosomes in the absence of Arl8.

5) In the experiments shown in Fig. 8C, the authors should determine the levels of LC3B-I and p62 in order to better evaluate the effect of DENND6A silencing on autophagic flux.

Response to reviewer 3, minor comment 5: We used a KO-validated LC3B antibody that strongly reacts with LC3B-II, the levels of which are normalized to the loading control for quantification. Despite high-exposure blots, we couldn't detect LC3B-I levels, as shown in the main figure. Similar patterns, mainly the LC3B-II band, were also observed in independent studies conducted by different research groups using the same antibody for quantification (references: PMID: 35314681, Fig. 8F-G; PMID: 33707434, Supp fig 2A-C; PMID: 34074205, Fig. 1B).

We thank the reviewer for suggesting the inclusion of p62 levels, a recommendation also made by reviewer 2. We have now added p62 blot in revised Fig. 8C and its quantification in revised Supp Fig. 14, suggesting reduced autophagy in the absence of DENND6A.

6) In the material and methods section, information on the antibody used in immunofluorescence for LC3 should be mentioned. Also, in the figure legends, what the yellow arrow depicts should be mentioned.

Response to reviewer 3, minor comment 6: We apologize for the oversight. We have now included information in the Materials and Methods section confirming that the same KO-validated LC3B antibody used for western blot analysis was also employed for immunofluorescence. Additionally, we have incorporated text into the figure legends to specifically reference the yellow arrows.

REVIEWER COMMENTS

Reviewer #1 (Remarks to the Author):

The additional experiments and modifications to the text have addressed some of my initial concerns. The proper analysis of co-localisation in Fig. 1C is obviously helpful (PCC now approx. 0.3).

I would still suggest that this is very low and the biological relevance is unclear, given the density of compartments in the juxtannuclear region of the cell. The new live-imaging in movie 4 reinforces by original concerns - I see little evidence that DENND6A-GFP and lysotracker are marking the same compartment, even at a low level.

One way to better support the authors' proposition would be to perform the additional (and very straightforward) analysis on the data associated with Figure 1C, where one extracts the juxtannuclear region of the cell in a tight square box, and rotates one channel by 90 degrees. If the PCC value reflects a biologically relevant correlation between the two signals rather than random/coincidental overlap, the PCC should drop significantly. Such an approach is described in detail in Dunn et al. (10.1152/ajpcell.00462.2010)

Reviewer #2 (Remarks to the Author):

This is a re-review of a manuscript by Kumar et al., where the authors examine the role that DENND6A plays in lysosomal positioning and function. The authors have done a good job in addressing my concerns and it is clear it represents a substantive body of work.

I still have some points about the autophagy experiments and the authors' interpretation of the results. While I do not propose additional experiments (they have already done a lot!), I suggest that the authors add a bit to the discussion to take the below into account.

The authors clearly demonstrate that loss of DENND6A dysregulates autophagy but given the change in expression levels, it may not be simply sufficient to say autophagy initiation is blocked. In figure 8, the

authors show reduced LC3II levels in KOs compared to WT. However, it still looks like the “relative flux” of LC3-II is the same for all cells. Under the EBSS conditions, the authors should plot WT +/- BAF together and KO +/- BAF together and compare significance within the genotype (as well as across it). In all cases it looks like EBSS reduces levels and BAF treatment restores levels back to those of untreated cells – i.e. autophagy flux. While it appears lower in the mutants, is this just because expression levels are lower? The p62 data supports this. Inhibition of autophagy should not lead to less p62 but more (as it is not degraded). Within each genotype BAF treatment does indeed lead to more p62. I think the authors need to consider that the lower expression of LC3 and p62 is not solely due to a block in autophagy but also a reduction in gene expression. Could the defect also be mTOR dysregulation rather than autophagy per se? mTOR is activated at lysosomes – and clearly starvation-induced dephosphorylation of the mTOR-dependent ULK1 phosphorylation is impaired (these are very nice data). This could mean that other mTOR substrates are not dephosphorylated – including TFEB. If TFEB remains phosphorylated it will not be able to translocate to the nucleus to activate transcription of lysosome/autophagy genes – which could explain the lower levels of LC3 and p62.

Reviewer #3 (Remarks to the Author):

The authors have satisfactorily addressed the comments.

The modifications are highlighted in **blue** in the revised manuscript and our detailed response (in **blue**) to the reviewer's comments can be found below:

Reviewer #1:

"The additional experiments and modifications to the text have addressed some of my initial concerns. The proper analysis of co-localisation in Fig. 1C is obviously helpful (PCC now approx. 0.3). I would still suggest that this is very low and the biological relevance is unclear, given the density of compartments in the juxtannuclear region of the cell. The new live-imaging in movie 4 reinforce by original concerns - I see little evidence that DENND6A-GFP and lysotracker are marking the same compartment, even at a low level".

"One way to better support the authors proposition would be to perform the additional (and very straightforward) analysis on the data associated with Figure 1C, where one extracts the juxtannuclear region of the cell in a tight square box, and rotates one channel by 90 degrees. If the PCC value reflects a biologically relevant correlation between the two signals rather than random/coincidental overlap, the PCC should drop significantly. Such an approach is described in detail in Dunn et al. (10.1152/ajpccell.00462.2010)".

Response to reviewer 1: We thank the reviewer for this suggestion. While we acknowledge that the PCC value of 0.3 is somewhat low, similar PCC values were reported between the Arl8 effector RUFY3 and LAMP1 (PMID- 35314681, Fig 1J; PMID- 35314674, Fig. 4B). We believe that we have justified the somewhat low PCC in the results section as per the reviewer's suggestion during the initial review.

Nevertheless, in response to this suggestion, we have now computed the PCC using cropped square-shaped regions from the juxtannuclear area in cells expressing either GFP or DENND6A-GFP, co-stained with LAMP1 (as illustrated in Figure 1B). Notably, we observed a substantial decrease in the PCC value upon a 90-degree rotation of one channel for DENND6A-GFP but not for GFP (Supp Fig. 1C of the revised manuscript).

Reviewer #2:

"This is a re-review of a manuscript by Kumar et al., where the authors examine the role that DENND6A plays in lysosomal positioning and function. The authors have done a good job in addressing my concerns and it is clear it represents a substantive body of work:.

"I still have some points about the autophagy experiments and the authors' interpretation of the results. While I do not propose additional experiments (they have already done a lot!), I suggest that the authors add a bit to the discussion to take the below into account".

"The authors clearly demonstrate a that loss of DENND6A dysregulates autophagy but given the change in expression levels, it may not be simply sufficient to say autophagy initiation is blocked. In figure 8, the authors show reduced LC3II levels in KOs compared to WT. However, it still looks like the "relative flux" of LC3-II is the same for all cells. Under the EBSS conditions, the authors should plot WT +/- BAF together and KO +/- BAF together and compare significance within the genotype (as well as across it). In all cases it looks like EBSS reduces levels and BAF treatment restores levels back to those of untreated cells – i.e. autophagy flux. While it appears lower in the mutants, is this just because expression levels are lower? The p62 data supports this. Inhibition of autophagy should not lead to less p62 but more (as it is

not degraded). Within each genotype BAF treatment does indeed lead to more p62. I think the authors need to consider that the lower expression of LC3 and p62 is not solely due to a block in autophagy but also a reduction in gene expression. Could the defect also be mTOR dysregulation rather than autophagy per se? mTOR is activated at lysosomes – and clearly starvation-induced dephosphorylation of the mTOR-dependent ULK1 phosphorylation is impaired (these are very nice data). This could mean that other mTOR substrates are not dephosphorylated – including TFEB. If TFEB remains phosphorylated it will not be able to translocate to the nucleus to activate transcription of lysosome/autophagy genes – which could explain the lower levels of LC3 and p62".

Response to reviewer 2: We are grateful for the valuable feedback provided by the reviewer, and we are pleased to learn that we have effectively addressed most of their previous concerns. We appreciate the reviewer's insightful input to incorporate additional possibilities into the discussion section, and we fully agree with their perspective. Following the reviewer's guidance, we have adjusted the presentation of Fig. 8D as to how we have grouped the phenotypes (Figure 1 in the response letter, below). While we acknowledge the reviewer's observation that the 'relative flux' appears similar across all conditions, our "group statistical analysis" revealed significance in the KO cells compared to the WT cells. However, within the KO groups, we observed variations in significance; for instance, unstarved/EBSS showed significance in KO2 but not in KO1, despite both displaying a similar trend. This discrepancy may be attributed to inherent cell-to-cell differences between the two KOs.

After careful consideration, we opted not to include this new data in the revised manuscript to prevent potential confusion among readers due to the significance variations within the KO groups. Nevertheless, we have fully integrated the interpretation of this data into the discussion section of the revised manuscript. In this section, we acknowledge the consistent trend in relative flux within all three conditions (WT/KO1/KO2) and speculate that the lower levels of LC3/p62 could be attributed to reduced expression of LC3/p62, possibly due to sustained phosphorylation of TFEB (page 15 of the revised manuscript).

Figure 1

Reviewer #3:

The authors have satisfactorily addressed the comments.

Response to reviewer 3: We are delighted to have successfully responded to the feedback provided by the reviewer.

REVIEWERS' COMMENTS

Reviewer #1 (Remarks to the Author):

The new data provided in S1C is helpful, thank you.

Despite some differences in interpretation, I think that this is an interesting paper, with very well performed experiments that offers some important new insights. As such, I am happy to recommend publication.

Reviewer #2 (Remarks to the Author):

The authors have now addressed my comments.